# The chemorepellent, SLIT2, bolsters innate immunity against *Staphylococcus aureus*

**Vikrant K Bhosle**[1], **Chunxiang Sun**[2], **Sajedabanu Patel**[1], **Tse Wing Winnie Ho**[3,4], **Johannes Westman**[1], **Dustin A Ammendolia**[1,5], **Fatemeh Mirshafiei Langari**[6,7], **Noah Fine**[2], **Nicole Toepfner**[8], **Zhubing Li**[1], **Manraj Sharma**[1], **Judah Glogauer**[1,2], **Mariana I Capurro**[1], **Nicola L Jones**[1,9,10,11], **Jason T Maynes**[6,12,13], **Warren L Lee**[3,4,7,14], **Michael Glogauer**[2,15,16], **Sergio Grinstein**[1,3,7], **Lisa A Robinson**[1,11,17,18]*

[1]Cell Biology Program, The Hospital for Sick Children Research Institute, Toronto, Canada; [2]Faculty of Dentistry, University of Toronto, Toronto, Canada; [3]The Keenan Research Centre for Biomedical Science, Unity Health Toronto, Toronto, Canada; [4]Department of Laboratory Medicine & Pathobiology, Medical Sciences Building, University of Toronto, Toronto, Canada; [5]Department of Molecular Genetics, Medical Sciences Building, University of Toronto, Toronto, Canada; [6]Program in Molecular Medicine, The Hospital for Sick Children Research Institute, Toronto, Canada; [7]Department of Biochemistry, Medical Sciences Building, University of Toronto, Toronto, Canada; [8]Department of Pediatrics, Faculty of Medicine and University Hospital Carl Gustav Carus, Technische Universität Dresden, Dresden, Germany; [9]Division of Gastroenterology, Hepatology and Nutrition, The Hospital for Sick Children, Toronto, Canada; [10]Department of Physiology, Medical Sciences Building, University of Toronto, Toronto, Canada; [11]Department of Paediatrics, Temerty Faculty of Medicine, University of Toronto, Toronto, Canada; [12]Department of Anesthesia and Pain Medicine, The Hospital for Sick Children, Toronto, Canada; [13]Department of Anesthesiology & Pain Medicine, Temerty Faculty of Medicine, University of Toronto, Toronto, Canada; [14]Department of Medicine and Interdepartmental Division of Critical Care Medicine, Temerty Faculty of Medicine, University of Toronto, Toronto, Canada; [15]Department of Dental Oncology and Maxillofacial Prosthetics, University Health Network, Princess Margaret Cancer Centre, Toronto, Canada; [16]Centre for Advanced Dental Research and Care, Mount Sinai Hospital, Toronto, Canada; [17]Institute of Medical Science, University of Toronto, Medical Sciences Building, University of Toronto, Toronto, Canada; [18]Division of Nephrology, The Hospital for Sick Children, Toronto, Canada

*For correspondence:
lisa.robinson@sickkids.ca

Competing interest: The authors declare that no competing interests exist.

**Abstract** Neutrophils are essential for host defense against *Staphylococcus aureus* (*S. aureus*). The neuro-repellent, SLIT2, potently inhibits neutrophil chemotaxis, and might, therefore, be expected to impair antibacterial responses. We report here that, unexpectedly, neutrophils exposed to the N-terminal SLIT2 (N-SLIT2) fragment kill extracellular *S. aureus* more efficiently. N-SLIT2 amplifies reactive oxygen species production in response to the bacteria by activating p38 mitogen-activated protein kinase that in turn phosphorylates NCF1, an essential subunit of the NADPH oxidase complex. N-SLIT2 also enhances the exocytosis of neutrophil secondary granules. In a murine model of *S. aureus* skin and soft tissue infection (SSTI), local SLIT2 levels fall initially but increase subsequently, peaking at 3 days after infection. Of note, the neutralization of endogenous

SLIT2 worsens SSTI. Temporal fluctuations in local SLIT2 levels may promote neutrophil recruitment and retention at the infection site and hasten bacterial clearance by augmenting neutrophil oxidative burst and degranulation. Collectively, these actions of SLIT2 coordinate innate immune responses to limit susceptibility to *S. aureus*.

## Editor's evaluation

Bhosle and colleagues present valuable findings on the function of the N-terminal fragment of SLIT2 in the amplification of reactive oxygen species production and exocytosis of secretory granules by neutrophils. The authors present solid in vitro and in vivo data supporting this unexpected role for SLIT2. This work advances our knowledge of innate immunity to pathogens.

## Introduction

*Staphylococcus aureus* (*S. aureus*) is a commensal bacterium as well as a skillful, facultative pathogen causing diverse human diseases ranging from localized SSTI to life-threatening disseminated sepsis (*Krismer et al., 2017*; *Lowy, 1998*). Neutrophils, the most abundant subset of leukocytes in circulation, form a formidable first-line of defense against *S. aureus* invasion and spread (*Guerra et al., 2017*). Accordingly, patients with neutropenia and defective neutrophil functions are highly susceptible to recurrent and more severe *S. aureus* infections (*Buvelot et al., 2017*; *Donadieu et al., 2011*; *Neehus et al., 2021*). Infections caused by antibiotic-resistant strains of *S. aureus* have been steadily increasing worldwide, suggesting that strategies to enhance the effective recruitment of neutrophils and their inherent bactericidal properties are needed to combat the morbidity and mortality associated with *S. aureus* infections (*Chambers and Deleo, 2009*; *Jernigan et al., 2020*).

A recent transcriptomic study noted that mRNA encoding Slit guidance ligand 2 (SLIT2), a canonical neuro-repellent, is locally upregulated during *S. aureus*-induced mastitis (*Günther et al., 2017*). Among the three mammalian SLIT family members, SLIT1 is exclusively expressed in the nervous system (*Wu et al., 2001*), while SLIT2 and SLIT3 are also detected outside the nervous system (*Bhosle et al., 2020*; *Kim et al., 2018*). We and others have previously reported that SLIT2 inhibits directed neutrophil migration in vitro as well as in vivo (*Chaturvedi et al., 2013*; *Tole et al., 2009*; *Wu et al., 2001*; *Ye et al., 2010*; *Zhou et al., 2022*). We also showed that the cleaved N-terminal fragment of SLIT2, N-SLIT2, acts via its receptor, Roundabout guidance receptor 1 (ROBO1), to attenuate inflammasome activation in macrophages by inhibiting macropinocytosis (*Bhosle et al., 2020*). Additionally, we and others previously reported that primary human and murine neutrophils express ROBO1 (*Rincón et al., 2018*; *Tole et al., 2009*; *Ye et al., 2010*), but not ROBO2 (*Rincón et al., 2018*). Despite the inhibitory actions of N-SLIT2 on chemotaxis and macropinocytosis in innate immune cells, administration of N-SLIT2 in vivo does not confer increased susceptibility to bacterial infections (*Chaturvedi et al., 2013*; *London et al., 2010*). It remains unknown if and how N-SLIT2 affects neutrophil responses during *S. aureus* infection.

Herein, we show that N-SLIT2 does not inhibit, but rather enhances, the killing of extracellular *S. aureus* by neutrophils. In the presence of N-SLIT2, human and murine neutrophils respond more robustly to *S. aureus* by activating p38 mitogen-activated protein kinase (MAPK) signaling, enhancing the production of extracellular reactive oxygen species (ROS) and release of secondary and tertiary granule contents by neutrophils. In a murine model of *S. aureus*-induced SSTI (*Prabhakara et al., 2013*), we found that endogenous levels of SLIT2 protein declined significantly early on but rose to peak levels approximately 3 days after infection, and that blocking endogenous SLIT2-ROBO1 signaling at the site of infection enhanced bacterial survival and worsened the infection. Our results suggest that changes in the levels of SLIT2 at local sites of infection may coordinate neutrophil recruitment, retention, and bactericidal responses to effectively and synergistically target *S. aureus*. Our work identifies SLIT2 as an endogenous regulator of neutrophil number and activity that coordinates immune responses vital to combat disseminated infection of bacterial pathogens.

## Results

### N-SLIT2 augments extracellular ROS production in response to *S. aureus*

We first investigated how SLIT2 affects neutrophil-mediated killing of *S. aureus*. Incubation of primary human neutrophils with bioactive N-SLIT2, but not inactive N-SLIT2ΔD2 which lacks the ROBO1/2-binding D2 LRR domain (*Patel et al., 2012*), significantly reduced extracellular *S. aureus* as early as 30 min (*Figure 1A*). Decreased extracellular bacteria could result from increased internalization via phagocytosis, and/or increased extracellular bacterial killing. Intriguingly, N-SLIT2 did not affect the ability of human neutrophils (*Figure 1B–D*) or RAW264.7 murine macrophages to phagocytose *S. aureus* (*Figure 1—figure supplement 1A–C*). To determine whether N-SLIT2 decreased bacterial survival by stimulating phagocyte ROS production, a process dependent on NADPH oxidase complex (NOX) activity, we incubated neutrophils with the pan-NOX-inhibitor, diphenyleneiodonium chloride (DPI). DPI partially restored extracellular bacterial survival in the presence of N-SLIT2 (*Figure 1—figure supplement 1D*). Next, we directly examined the effects of N-SLIT2 on extracellular ROS production by human neutrophils exposed to *S. aureus*. Neutrophils incubated with vehicle control or with N-SLIT2 alone had very low basal levels of ROS. As expected, *S. aureus* induced significant ROS production that was indistinguishable from that seen in cells incubated with *S. aureus* and bio-inactive N-SLIT2ΔD2 (*Figure 1E–F*). Surprisingly, human neutrophils co-incubated with bio-active N-SLIT2 and *S. aureus* produced significantly more extracellular ROS than *S.aureus*-exposed neutrophils incubated with vehicle or N-SLIT2ΔD2 (*Figure 1E–F*). We found that neutrophils exposed to another secondary ROS-inducing stimulus, namely phorbol-12-myristate-13-acetate (PMA), also produced more extracellular ROS in the presence of N-SLIT2 compared to N-SLIT2ΔD2 (*Figure 1—figure supplement 1E*). Similar observations were also noted for mouse bone marrow-derived neutrophils (BMDN) indicating that the effect of N-SLIT2 to further enhance extracellular ROS production by *S. aureus*-exposed neutrophils is not species-specific (*Figure 1—figure supplement 1F*). To determine whether the observed effects of N-SLIT2 occurred through the canonical ROBO1 receptor, we pre-incubated N-SLIT2 with the soluble N-terminal fragment of the ROBO1 receptor (N-ROBO1). N-ROBO1 fully blocked the ability of N-SLIT2 to boost extracellular ROS production in neutrophils exposed to *S. aureus* (*Figure 1G–H*). Since activation of Rac GTPases is essential for optimal NOX function in neutrophils (*Diebold and Bokoch, 2001*; *Hordijk, 2006*), we next tested the effects of N-SLIT2 on Rac. N-SLIT2 alone failed to activate Rac but instead required a second stimulus, namely *S. aureus* (*Figure 1—figure supplement 1G*). Together, these results demonstrate that N-SLIT2 enhances neutrophil-mediated killing of *S. aureus*, partly by amplifying extracellular ROS production in a ROBO1-dependent manner.

### N-SLIT2 primes NOX by p38-mediated phosphorylation of NCF1

We next studied how N-SLIT2 increases extracellular ROS production in neutrophils. We wondered whether N-SLIT2 induces NOX priming and extracellular ROS production by prompting phosphorylation and translocation of Neutrophil Cytosolic Factor 1 (NCF1; p47phox), a key component of the NOX complex, to the plasma membrane (*El-Benna et al., 2009*; *Li et al., 2010*). We found that N-SLIT2 increased phosphorylation of the conserved $Ser^{345}$ residue of NCF1 in neutrophils (*Figure 2A–B*), as well as in RAW264.7 macrophages (*Figure 2—figure supplement 1A–B*). As SLIT2-ROBO2 signaling was recently reported to activate NOX function in neurons by activating protein kinase C (PKC) (*Terzi et al., 2021*), we examined the effects of N-SLIT2 on PKC activation in neutrophils. PKC activity in cells is tightly regulated by its phosphorylation (*Freeley et al., 2011*). N-SLIT2 did not induce phosphorylation of PKC in neutrophils (*Figure 2—figure supplement 1C–F*). SLIT2 orthologues have been reported to activate p38 MAPK signaling in *Xenopus* and *C. elegans* neurons (*Campbell and Okamoto, 2013*; *Piper et al., 2006*). The p38 MAP kinases are also known to phosphorylate NCF1 at the $Ser^{345}$ residue to mediate NOX priming (*Dang et al., 2006*). We, therefore, investigated the effects of N-SLIT2 on p38 MAPK signaling in mammalian phagocytes. N-SLIT2 potently induced p38 MAPK activation in both neutrophils (*Figure 2C–D*) and macrophages (*Figure 2—figure supplement 1G–H*). Congruent with these results, incubation with two distinct pharmacologic inhibitors of p38 MAPKs blocked the observed N-SLIT2-induced increase in extracellular ROS in neutrophils exposed to *S. aureus* (*Figure 2E–F*). We and others previously showed that SLIT2-ROBO1 signaling robustly activates RhoA, which in turn can activate Rho-associated protein kinases (ROCK), in several cell types, including macrophages and cancer cells (*Bhosle et al., 2020*; *Kong et al., 2015*). We found that in the

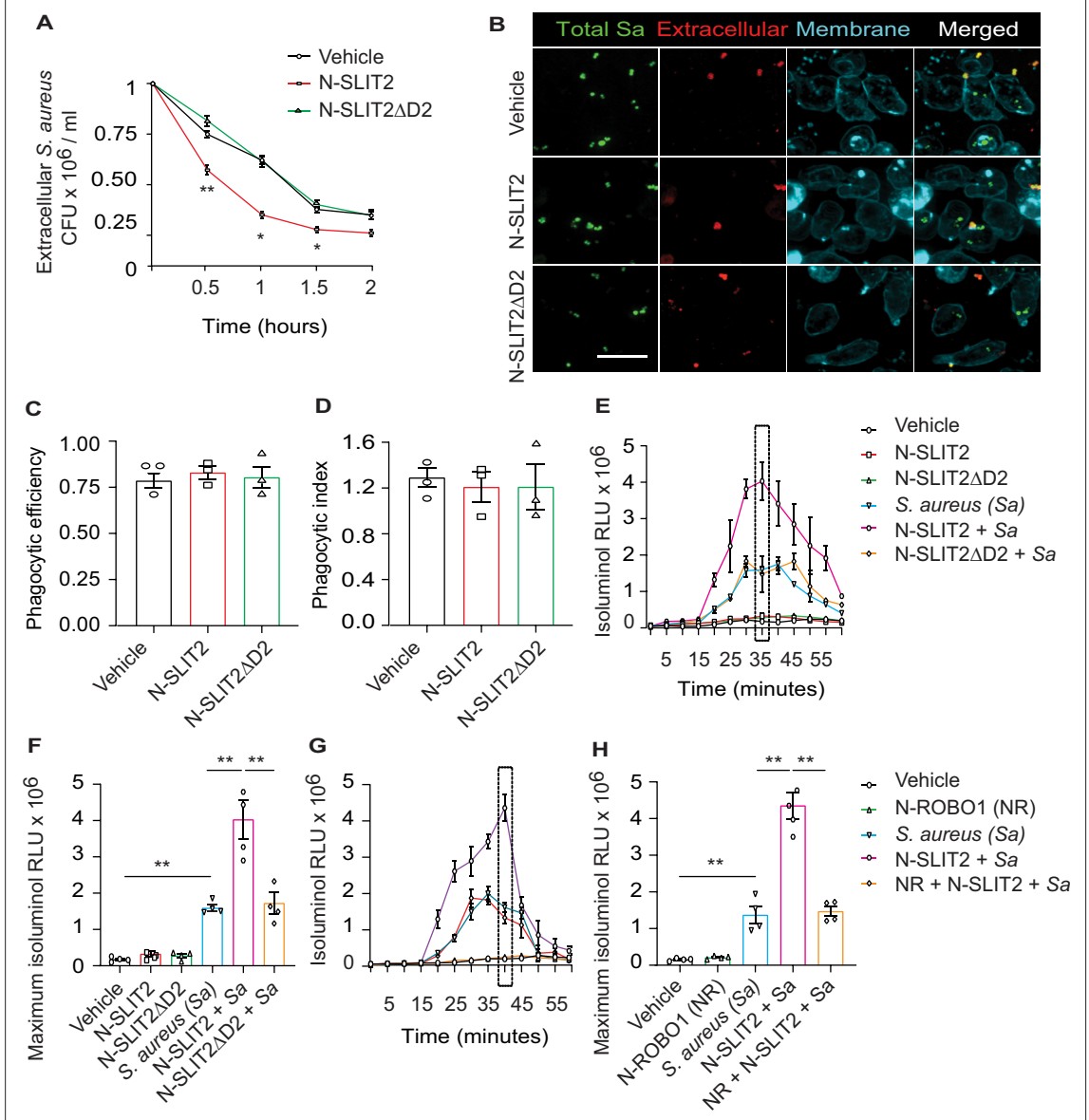

**Figure 1.** N-SLIT2 augments extracellular reactive oxygen species (ROS) production in response to *S. aureus*. (**A**) Neutrophils, isolated from healthy human donors, were incubated with vehicle (HBSS), N-SLIT2 (30 nM) or N-SLIT2ΔD2 (30 nM) for 15 min, followed by exposure to *S. aureus* (MOI 10) for the indicated times. Extracellular *S. aureus* counts were determined by serial dilution. n=3. The statistical comparisons between N-SLIT2 and N-SLIT2ΔD2 groups are shown. p=0.0072 (0.5 hr), p=0.0105 (1 hr), p=0.0478 (1.5 hr), and p=0.0852 (2 hr). (**B**) Human neutrophils were treated with vehicle, N-SLIT2 or N-SLIT2ΔD2 for 15 min and then incubated with unoposnized *S. aureus* expressing GFP (MOI 10) for an additional 45 min. Extracellular bacteria were labeled using donkey anti-human IgG-Cy3. Neutrophil plasma membranes were labeled using Concanavalin A-AF647. At least 100 neutrophils per treatment were imaged. n=3. The phagocytic efficiency (**C**) and index (**D**) were calculated. (**E**) The experiments were performed as in 'A' and extracellular ROS production was measured every 5 min using isoluminol relative luminescent units (RLU). n=4. The averages of four experiments are shown. The timepoint with maximum extracellular ROS (35 min) is marked with a dotted rectangle. (**F**) Extracellular ROS production corresponding to maximum isoluminol RLU was compared among experimental groups. p=0.0031 (vehicle vs *S. aureus*), p=0.0099 (*S. aureus* vs N-SLIT2 + *S. aureus*), and p=0.0055 (N-SLIT2 + *S. aureus* vs N-SLIT2ΔD2 + *S. aureus*). (**G**) Experiments were performed as described In (**E**) in parallel incubating N-SLIT2 (30 nM) with N-ROBO1 (NR; 90 nM) for 1 hr at room temperature before adding to the cells. n=4. Averages of all experiments are shown. The timepoint with maximum extracellular ROS (40 min) is marked with a dotted rectangle. (**H**) The timepoint with maximum isoluminol relative luminescent units (RLU) was compared across experimental groups. p=0.0057 (vehicle vs *S. aureus*), p=0.0018 (*S. aureus* vs N-SLIT2 + *S. aureus*), and p=0.0028 (N-SLIT2 + *S. aureus* vs N-ROBO1 +N-SLIT2 + *S. aureus*). Mean values ± SEM. *p<0.05, and **p<0.01. The source data are available as *Figure 1—source data 1*.

The online version of this article includes the following source data and figure supplement(s) for figure 1:

**Source data 1.** The file contains source data for *Figure 1A, C, D, F, H*.

*Figure 1 continued on next page*

Figure 1 continued

**Figure supplement 1.** Anti-staphylococcal actions of N-SLIT2 are partially dependent on increases in extracellular ROS production.

**Figure supplement 1—source data 1.** The file contains source data for *Figure 1—figure supplement 1B–G*.

presence of the selective ROCK inhibitor, Y-27632, N-SLIT2 failed to activate p38 MAPK in neutrophils (*Figure 2G–H*). These findings indicate that N-SLIT2 does not activate, but rather primes, the NOX complex in a p38 MAPK-dependent manner in phagocytes to upregulate ROS production in response to injurious biologic and pharmacological secondary stimuli, including *S. aureus* and PMA, respectively. Additionally, N-SLIT2-induced activation of the RhoA/ROCK pathway is essential for its effect on p38 MAPK signaling in neutrophils.

## N-SLIT2 enhances p38 MAPK-mediated exocytosis of secondary and tertiary granules

Sequential cytoskeletal changes in neutrophils have been shown to play an important role in their priming (*Bashant et al., 2019*; *Toepfner et al., 2018*). We performed Real-time deformability cytometry (RT-DC) and found that exposure of blood to N-SLIT2 reduced neutrophil cell area and deformability although the effect did not reach statistical significance (*Figure 3—figure supplement 1A–B*; *Bashant et al., 2019*; *Toepfner et al., 2018*). Next, since exposure to the NOX inhibitor, DPI, only partially reversed the effects of N-SLIT2 on extracellular *S. aureus* survival, we pondered whether other anti-microbial functions of neutrophils are also modified by N-SLIT2. In addition to the oxidative burst, degranulation has been shown to play an important role in the bactericidal responses of neutrophils against *S. aureus* (*Ferrante et al., 1989*; *Van Ziffle and Lowell, 2009*). Neutrophils contain four types of granules, each containing different classes of anti-microbial peptides, of which secondary and tertiary granules are most important for eliminating *S. aureus* (*Borregaard and Cowland, 1997*; *Mollinedo, 2019*; *Van Ziffle and Lowell, 2009*). To investigate the effects of N-SLIT2 on degranulation, we utilized a recently optimized flow cytometry protocol using specific cluster of differentiation (CD) markers of granule exocytosis (*Fine et al., 2019*). Another advantage of using a flow cytometry-based approach is that it circumvents the in vitro neutrophil isolation step which is known to affect cellular activation (neutrophil gating strategy; *Figure 3A*). Neither *S. aureus* alone nor *S. aureus* together with N-SLIT2 induced primary granule (CD63) secretion (*Figure 3—figure supplement 1C*). Congruent with published literature (*Lu et al., 2014*; *Schmidt et al., 2012*), the bacteria alone stimulated secondary granule (CD66b) exocytosis from neutrophils (*Figure 3—figure supplement 1D*). This effect was strikingly enhanced by exposure to N-SLIT2 (*Figure 3B* and *Figure 3—figure supplement 1D*). Additionally, the actions of N-SLIT2 were completely obliterated by pharmacological inhibition of p38 MAPK signaling using SB 203580 or p38 MAPK Inhibitor IV, but not MEK1/2 signaling using PD 184161 (*Figure 3B*). It is noteworthy that N-SLIT2 also augmented surface expression of *S. aureus*-induced CD18, which is stored in both secondary and tertiary granules, in a p38 MAPK dependent manner (*Figure 3C* and *Figure 3—figure supplement 1E*; *Borregaard and Cowland, 1997*; *Mollinedo, 2019*). On the other hand, we found no differences in the surface expression of CD16, which is stored in secretory vesicles, in any of the tested conditions, a finding in agreement with the priming-associated CD marker signature described for neutrophils in circulation and tissues (*Figure 3—figure supplement 1F*; *Fine et al., 2019*). We next used the surface expression of CD66b and CD11b to calculate the percentage of primed neutrophils (*Figure 3—figure supplement 1D and G*; *Fine et al., 2019*). Exposure of blood to N-SLIT2 and *S. aureus* together significantly increased the fraction of primed neutrophils as compared to *S. aureus* alone (*Figure 3D*). Next, we measured the secretion of LL-37, a highly potent cationic anti-staphylococcal peptide that is selectively stored in neutrophil secondary granules in its pro-peptide form (*Noore et al., 2013*; *Sørensen et al., 1997*). Incubation of human neutrophils with *S. aureus* resulted in the release of LL-37, which was further amplified in the presence of N-SLIT2 and blocked by the p38 inhibitors (*Figure 3E*). Production of neutrophil extracellular traps (NETosis) is known to be modulated by intra- and extra-cellular ROS and can also regulate anti-bacterial responses by neutrophils in vitro and in vivo (*Poli and Zanoni, 2023*). We investigated the effects of N-SLIT2 on NETosis and found that exposure to N-SLIT2 alone failed to induce NETosis. Treatment of neutrophils with N-SLIT2 and *S. aureus* together resulted in more NETosis than that with the bacteria alone but this effect was not statistically significant (*Figure 3—figure supplement 1H–J*).

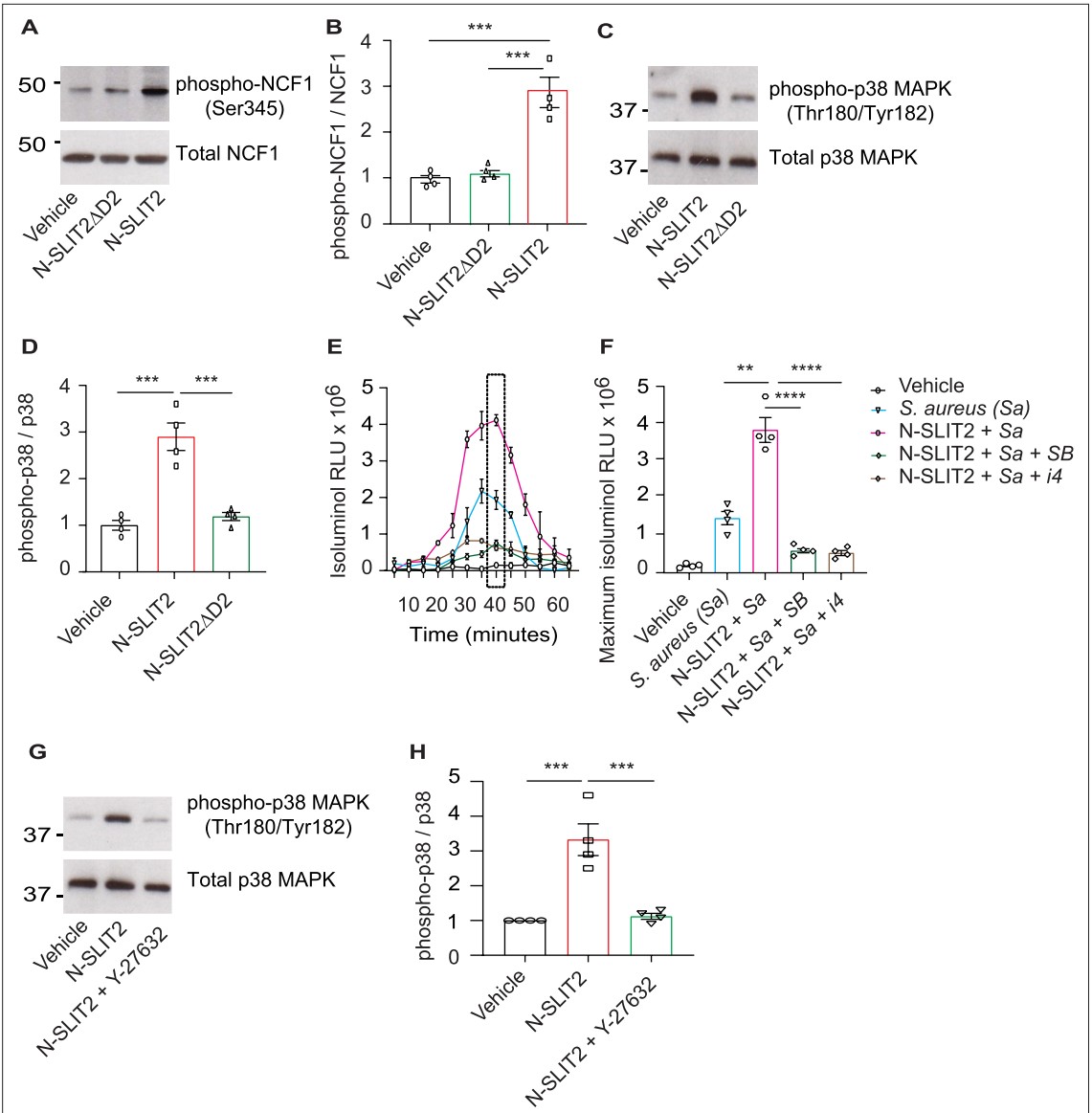

**Figure 2.** N-SLIT2 primes NADPH oxidase complex (NOX) by p38-mediated phosphorylation of Neutrophil Cytosolic Factor 1 (NCF1). (**A**) Human neutrophils were exposed to vehicle, N-SLIT2 or N-SLIT2ΔD2 for 15 min and the protein lysates were immunoblotted for phospho-NCF1 (Ser345) and total NCF1 (Ser345). n=4. A representative blot is shown. (**B**) Experiments were performed as in (**A**), densitometry was performed, and the ratio of phospho-NCF1 /NCF1 was obtained. n=4. p=0.0002 (vehicle vs N-SLIT2) and p=0.0004 (N-SLIT2 vs N-SLIT2ΔD2). (**C**) Experiments were performed as in (**A**) and immunoblotting performed for phospho-p38 (Thr180/Tyr182) and total p38. n=4. A representative blot is shown. (**D**) Experiments were performed as in (**C**), densitometry was performed, and the ratio of phospho-p38 /p38 was obtained. p=0.0006 (vehicle vs N-SLIT2) and p=0.0004 (N-SLIT2 vs N-SLIT2ΔD2). (**E**) Human neutrophils were incubated with vehicle, *S. aureus*, N-SLIT2 + *S. aureus*, N-SLIT2 + *S. aureus* + SB203580 (SB, 10 μM), or N-SLIT2 + *S. aureus* + p38 MAPK Inhibitor IV (i4, 10 μM), and extracellular reactive oxygen species (ROS) were measured as described in Figure (1E). n=4. The averages of four experiments are shown. The timepoint with maximum extracellular ROS (40 min) is marked with a dotted rectangle. (**F**) Experiments were performed as in (**E**) and extracellular ROS production at 40 min compared among groups. n=4. p=0.0036 (*S. aureus* vs N-SLIT2 + *S. aureus*), p<0.0001 (N-SLIT2 + *S. aureus* vs N-SLIT2 + *S. aureus* +SB203580), and p<0.0001 (N-SLIT2 + *S. aureus* vs N-SLIT2 + *S. aureus* + p38 MAPK Inhibitor IV). (**G**) Human neutrophils were exposed to vehicle, N-SLIT2 or N-SLIT2 and Y-27632 together for 15 min and immunoblotting was performed for phospho-p38 (Thr180/Tyr182) and total p38. n=4. A representative blot is shown. (**H**) Experiments were performed as in (**G**), densitometry was performed, and the ratio of phospho-p38 /p38 was obtained. p=0.0004 (vehicle vs N-SLIT2) and p=0.0007 (N-SLIT2 vs N-SLIT2 + Y-27632). Mean values ± SEM. **p<0.01, ***p<0.001, and ****p<0.0001. The source data are available as *Figure 2—source data 1* and *Figure 2—source data 2*.

The online version of this article includes the following source data and figure supplement(s) for figure 2:

**Source data 1.** The file contains source data for *Figure 2B, D, F, H*.

**Source data 2.** The file contains source data for *Figure 2A, C, G*.

*Figure 2 continued on next page*

*Figure 2 continued*

**Figure supplement 1.** N-SLIT2 does not activate PKC signaling in neutrophils.

**Figure supplement 1—source data 1.** The file contains source data for *Figure 2—figure supplement 1B, D, F, H*.

**Figure supplement 1—source data 2.** The file contains source data for *Figure 2—figure supplement 1A, C, E, G*.

Collectively, our results indicate that SLIT2-induced activation of p38 MAPK augments the production of extracellular ROS, exocytosis of secondary granules, and secretion of the anti-bacterial LL-37 peptide from neutrophils in response to exposure to *S. aureus*.

## Blocking endogenous SLIT2 exacerbates tissue injury in *S. aureus* SSTI

A recent transcriptomic screen has reported that SLIT2 mRNA is upregulated during *S. aureus*-induced mastitis (*Günther et al., 2017*). To determine whether this increase is conserved at a protein level during SSTI, we used an established murine model of *S. aureus* SSTI, which mimics community-acquired human infections (*Prabhakara et al., 2013*). Surprisingly, we found that local SLIT2 levels were reduced as early as 12 hr after the infection (*Figure 4A*). Following the initial decrease, the levels of SLIT2 gradually increased to reach a peak at 3 days (*Figure 4A*). Unlike SLIT2, SLIT3 levels in the infected tissue remained significantly lower than in control mock-infected tissue for the first 2 days (*Figure 4—figure supplement 1A*). To ascertain the target of SLIT2 in *S. aureus* SSTI, we used a soluble N-ROBO1 to block endogenous SLIT2 (*Bhosle et al., 2020*; *Geraldo et al., 2021*). Based on the observed peak in endogenous SLIT2 levels at Day 3 after *S. aureus* infection, N-ROBO1 or control IgG was administered on Days 2 and 3 after infection (*Figure 4—figure supplement 1B*). Strikingly, the bacterial counts were more than twofold higher in N-ROBO1-treated mice as compared to the IgG-treated counterparts (*Figure 4—figure supplement 1C*). We and others have shown that SLIT2-ROBO1 signaling inhibits chemotactic neutrophil migration in vivo but this has not been examined in the context of an infection so far (*Chaturvedi et al., 2013*; *Tole et al., 2009*; *Zhou et al., 2022*). *S. aureus* infection in mice who received control IgG treatment or no other treatment resulted in microabscess formation with immune cell infiltration (*Figure 4B–D*). In N-ROBO1-treated mice, *S. aureus* infection caused much more extensive tissue injury characterized by diffuse, rather than localized, inflammation (*Figure 4B–D*). Interestingly, in the absence of bacterial infection, neutralization of endogenous SLIT2 augmented immune cell infiltration in the skin but did not result in local tissue damage in the form of acanthosis (*Figure 4C*). Next, we directly examined tissue neutrophil infiltration in SSTI using immunohistochemical (IHC) staining (Ly6G⁺F4/80⁻ cells) (*Chadwick et al., 2021*). In the absence of SSTI, N-ROBO1 treatment alone was sufficient to increase neutrophil infiltration (*Figure 4—figure supplement 1D–E*). In line with our earlier findings using H&E, animals administered *S. aureus* and N-ROBO1 exhibited significantly more local neutrophil infiltration in infected tissue as compared to those administered *S. aureus* alone or *S. aureus* with control IgG (*Figure 4—figure supplement 1D–E*). Finally, we used IHC to measure 8-hydroxy-2'-deoxyguanosine (8-OHdG) levels as a marker of free radical-induced oxidative tissue injury (*Sima et al., 2016*). N-ROBO1-treated mice had an almost 50% reduction in the 8-OHdG-positive lesion area as compared to animals which received either *S. aureus* alone or *S. aureus* and control IgG (*Figure 4—figure supplement 1F–G*). Together, our findings suggest that immediately after infection the rapid decrease in local levels of SLIT2 promotes infiltration of neutrophils into the site of infection. The later increase in SLIT2 levels may serve dual functions: local retention of neutrophils due to SLIT2's chemorepellent actions, and direct effects on neutrophils to enhance ROS production and anti-staphylococcal killing responses.

## Dermal microvascular endothelial cells are a source of SLIT2 during *S. aureus* infection

We next investigated the potential cellular source of SLIT2 during SSTI. Under specific non-infectious conditions, dermal microvascular endothelial cells (DMEC) and fibroblasts have been previously reported to secrete SLIT2 (*Pilling et al., 2014*; *Romano et al., 2018*). To test this possibility, we used a human dermal microvascular endothelial cell line, HMEC-1 (*Ades et al., 1992*). Infection of HMEC-1 with *S. aureus* was sufficient to reduce SLIT2 production at 12 hr at the level of mRNA as well as protein (*Figure 5A–B*). However, the effect was reversed at 48 hr, at which time infected cells produced more SLIT2 than their non-infected counterparts (*Figure 5A–B*). We next examined how

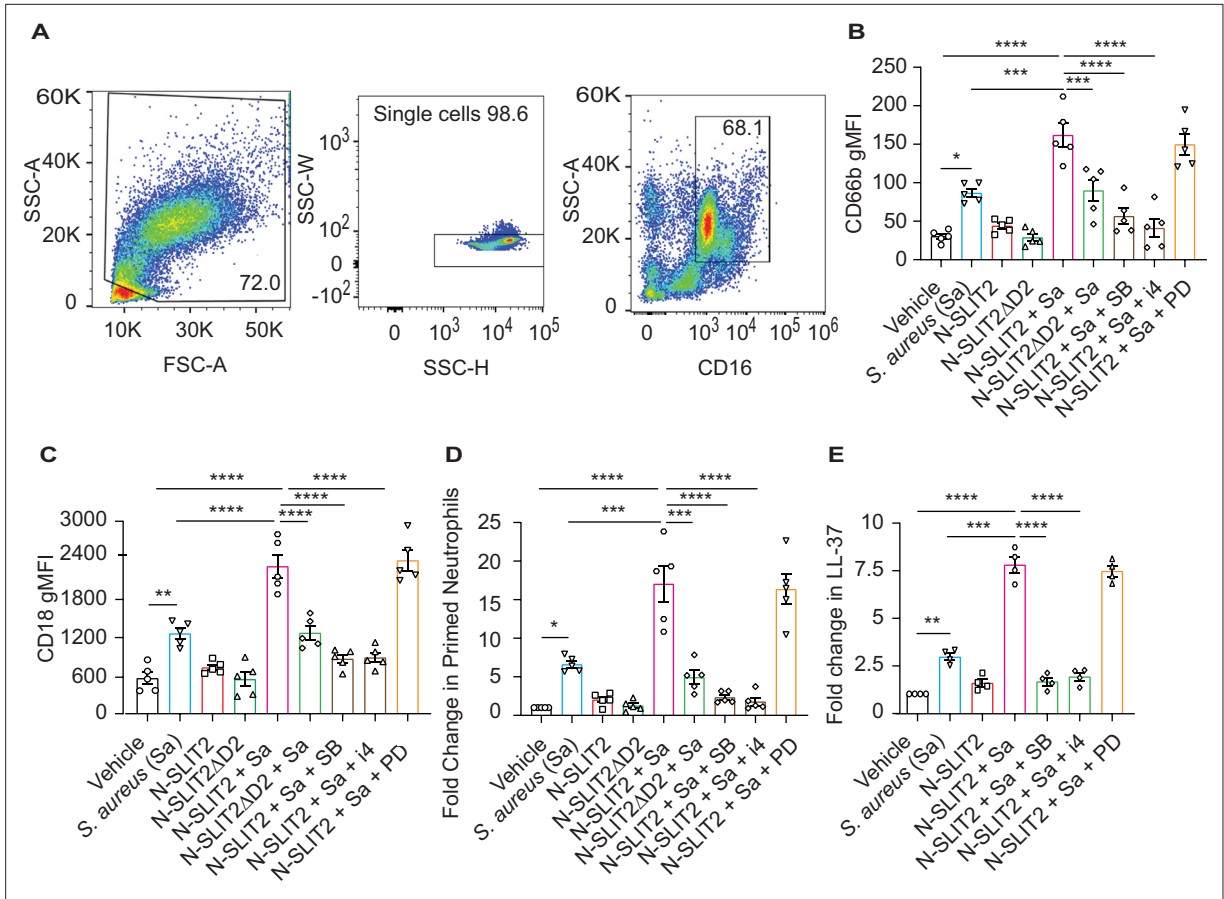

**Figure 3.** N-SLIT2 enhances p38 MAPK-mediated exocytosis of secondary and tertiary granules. (**A–D**) 100 µl whole blood from human subjects was exposed to different treatments for 15 min at 37 °C, as indicated. The samples were immediately fixed on ice with 1.6% paraformaldehyde (PFA) and surface CD markers labeled. n=5. (**A**) Gating strategy for human blood neutrophils: Red blood cells and dead cell debris were excluded based on FSC-A × SSC-A. Doublets were excluded based on SSC-A × SSC-W. Neutrophils were gated in whole blood leukocytes using CD16high × SSC-Ahigh. (**B**) Human neutrophils were exposed to vehicle, N-SLIT2, or N-SLIT2ΔD2 with or without the p38 MAPK inhibitors, SB 203580 (SB; 10 µM) or p38 MAPK Inhibitor IV (i4; 10 µM), or the MEK1/2 inhibitor PD 184161 (PD; 10 µM) for 15 min, followed by exposure to *S. aureus* (*Sa*) for another 15 min at 37 °C, as indicated. Geometric mean fluorescence intensity (gMFI) for CD66b (secondary granules) is shown. p=0.0122 (vehicle vs *Sa*), p<0.0001 (vehicle vs N-SLIT2 + *Sa*) p=0.0003 (*Sa* vs N-SLIT2 + *Sa*), p=0.0006 (N-SLIT2 + *Sa* vs N-SLIT2ΔD2 + *Sa*), p<0.0001 (N-SLIT2 + *Sa* vs N-SLIT2 + *Sa* + SB), and p<0.0001 (N-SLIT2 + *Sa* vs N-SLIT2 + *Sa* + i4). (**C**) Neutrophils were treated as in (**B**) and gMFI for CD18 (secondary and tertiary granules) is noted. p=0.0022 (vehicle vs *Sa*), p<0.0001 (vehicle vs N-SLIT2 + *Sa*) p<0.0001 (*Sa* vs N-SLIT2 + *Sa*), p<0.0001 (N-SLIT2 + *Sa* vs N-SLIT2ΔD2 + *Sa*), p<0.0001 (N-SLIT2 + *Sa* vs N-SLIT2 + *Sa* + SB), and p<0.0001 (N-SLIT2 + *Sa* vs N-SLIT2 + *Sa* + i4). (**D**) Human neutrophils were exposed to vehicle, N-SLIT2, or N-SLIT2ΔD2 with or without the p38 MAPK inhibitors, SB 203580 (SB; 10 µM) or p38 MAPK Inhibitor IV (i4; 10 µM), or the MEK1/2 inhibitor PD 184161 (PD; 10 µM) for 15 min, followed by exposure to *S. aureus* (*Sa*) for another 15 min at 37 °C, as indicated. Primed neutrophils were identified by cell surface labeling CD66bhigh × CD11bhigh and fold changes in % primed neutrophils relative to vehicle treatment are shown. p=0.0246 (vehicle vs *Sa*), p<0.0001 (vehicle vs N-SLIT2 + *Sa*) p=0.0002 (*Sa* vs N-SLIT2 +*Sa*), p=0.0008 (N-SLIT2 + *Sa* vs N-SLIT2ΔD2+ *Sa*), p<0.0001 (N-SLIT2 + *Sa* vs N-SLIT2 + *Sa* + SB), and p<0.0001 (N-SLIT2 + *Sa* vs N-SLIT2 + *Sa* + i4). (**E**) Human neutrophils were exposed to vehicle or N-SLIT2 with or without p38 MAPK inhibitors, SB or i4, or the MEK1/2 inhibitor PD for 15 min, then exposed to *S. aureus* (*Sa*) for another 15 min at 37 °C, as indicated. Supernatants were collected and secreted LL-37 levels were measured using an ELISA. n=4. p=0.0092 (vehicle vs *Sa*), p<0.0001 (vehicle vs N-SLIT2 + *Sa*) p=0.0005 (*Sa* vs N-SLIT2 + *Sa*), p<0.0001 (N-SLIT2 + *Sa* vs N-SLIT2 + *Sa* + SB), and p<0.0001 (N-SLIT2 + *Sa* vs N-SLIT2 + *Sa* + i4). Mean values ± SEM. *p<0.05, **p<0.01, ***p<0.001 and ****p<0.0001. The source data are available as *Figure 3—source data 1*.

The online version of this article includes the following source data and figure supplement(s) for figure 3:

**Source data 1.** The file contains source data for *Figure 3B–E*.

**Figure supplement 1.** N-SLIT2 does not significantly affect NETosis.

**Figure supplement 1—source data 1.** The file contains source data for *Figure 3—figure supplement 1A, B, I, J*.

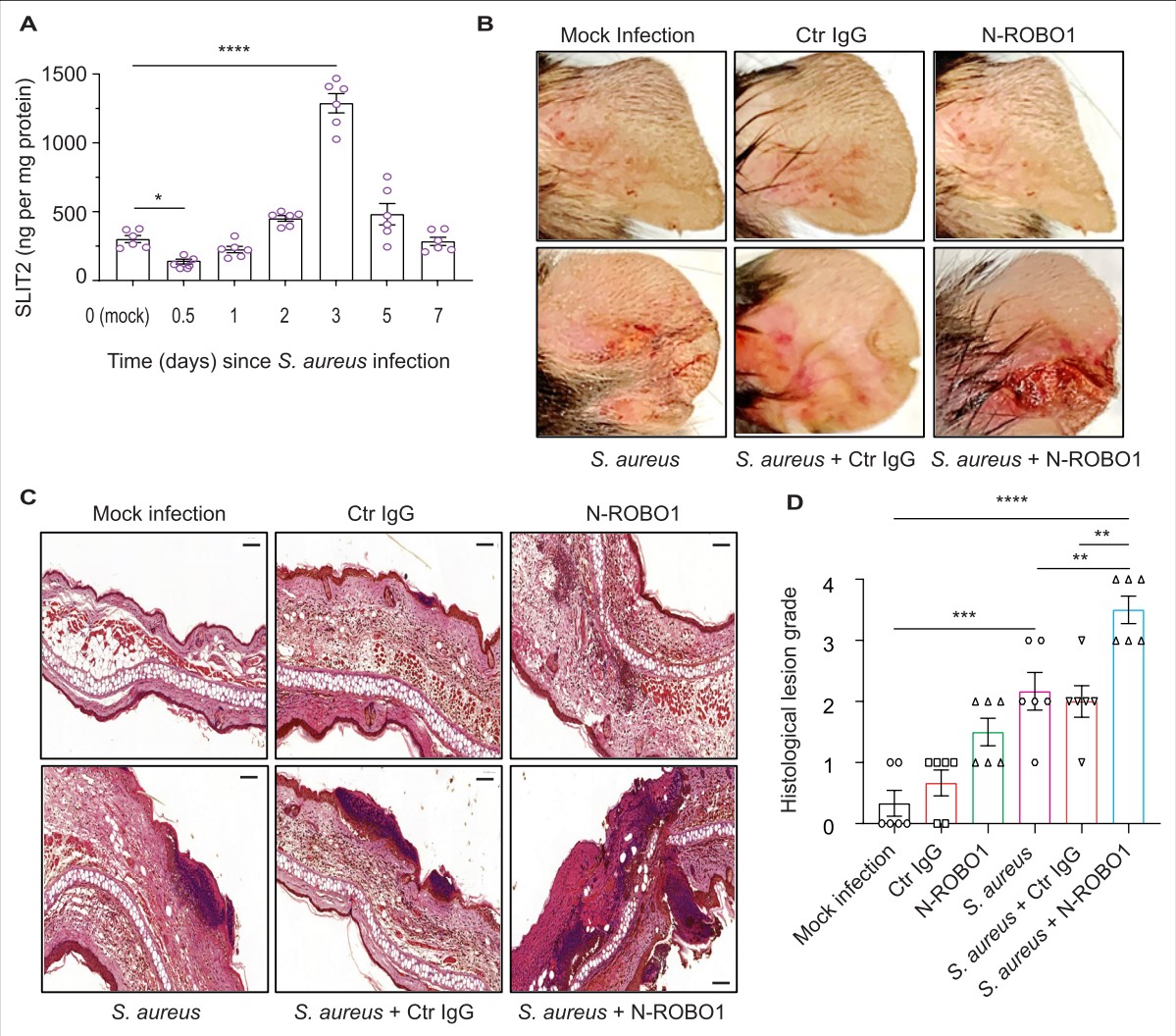

**Figure 4.** Blocking endogenous SLIT2 exacerbates tissue injury in *S. aureus* skin and soft tissue infection (SSTI). (**A**) Ear skin samples were collected from mock-infected (0) and *S. aureus*-infected mice at indicated time points, homogenized, and tissue SLIT2 levels were measured using an ELISA. n=6 mice per group. p=0.0290 (Mock infection vs *S. aureus* 0.5 day), p<0.0001 (Mock infection vs *S. aureus* 3 days). (**B**) Representative images of gross pathology of ear tissue from animals treated as described in (*Figure 4—figure supplement 1B*). (**C–D**) Samples were collected as described in (**B**), fixed in formalin, and stained with hematoxylin and eosin. Scale bar = 100 μm (**D**) Experiments were performed as in (**C**). The lesions were blindly scored on an ascending scale of severity (0–5). n=6. p=0.0002 (Mock infection vs *S. aureus*), p<0.0001 (Mock infection vs *S. aureus* + N-ROBO1), p=0.0060 (*S. aureus* vs *S. aureus* + N-ROBO1), p=0.0016 (*S. aureus* + Ctr IgG vs *S. aureus* + N-ROBO1). Mean values ± SEM. *p<0.05, **p<0.01, ***p<0.001, and ****p<0.0001. The source data are available as *Figure 4—source data 1*.

The online version of this article includes the following source data and figure supplement(s) for figure 4:

**Source data 1.** The file contains source data for *Figure 4A, D*.

**Figure supplement 1.** Inhibition of endogenous SLIT2 signaling increases tissue neutrophil infiltration, and attenuates tissue-associated ROS.

**Figure supplement 1—source data 1.** The file contains source data for *Figure 4—figure supplement 1A, C, E, G*.

other factors which could be found in a Staphylococcal abscess, namely hypoxia and low pH (acidosis), (*Costa and Horswill, 2022*; *Zhang et al., 2022*), modulate SLIT2 production by vascular endothelial cells. Lowering the pH of the medium to 6.6 did not change *SLIT2* mRNA in HMEC-1 cells (*Figure 5—figure supplement 1A*). In contrast, growing the cells in a hypoxic (1% O$_2$) environment replicated the effects of *S. aureus* infection on SLIT2 production (*Figure 5C–D*). Finally, we tested the ability of conditioned media from HMEC-1 cells to regulate LL-37 secretion by neutrophils. Conditioned media from HMEC-1 cells taken 48 hr after bacterial infection boosted LL-37 secretion and this effect was blocked by SLIT2 neutralization using N-ROBO1 (*Figure 5—figure supplement 1B*). The same effect

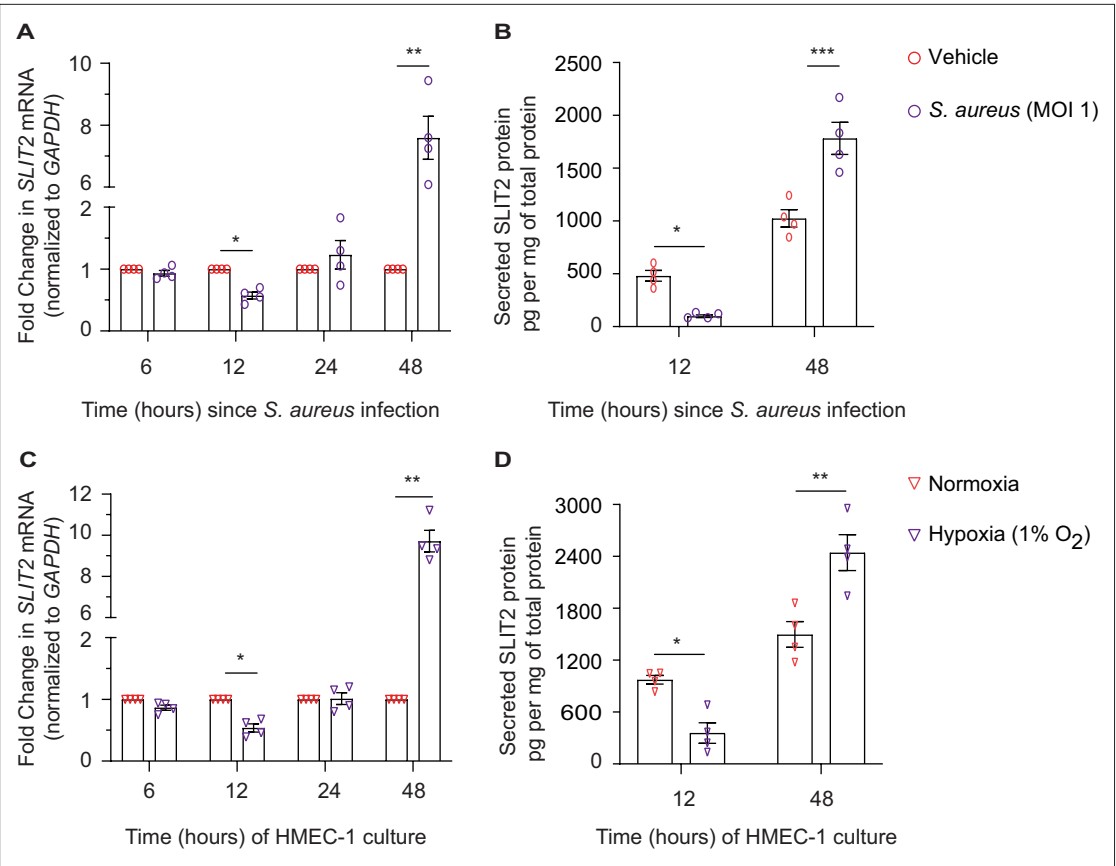

**Figure 5.** Dermal microvascular endothelial cells are a source of SLIT2 during *S. aureus* infection. (A–B) HMEC-1 cells were infected with *S. aureus* (MOI 1), where indicated and cultured for 6, 12, 24, and 48 hr. Total cellular RNA and culture media were collected at each time point. n=4 biological replicates. (A) *SLIT2* and *GAPDH* (housekeeping gene) mRNA expression levels were measured using Quantitative PCR (qPCR). p=0.0191 (vehicle vs *S. aureus* at 12 hr), and p=0.0100 (vehicle vs *S. aureus* at 48 hr). (B) SLIT2 levels were measured in the culture supernatant using ELISA and normalized to the total protein concentration in the supernatant. p=0.0222 (vehicle vs *S. aureus* at 12 hr), and p=0.0001 (vehicle vs *S. aureus* at 48 hr). (C–D) HMEC-1 cells were cultured in normoxic (20% $O_2$) or hypoxic (1% $O_2$) conditions for 6, 12, 24, and 48 hr, as indicated. Total cellular RNA and culture media were collected at each time point. n=4 biological replicates. (C) *SLIT2* and *GAPDH* (housekeeping gene) mRNA expression levels were measured using qPCR. p=0.0253 (normoxia vs hypoxia at 12 hr), and p=0.0019 (normoxia vs hypoxia at 48 hr). (D) SLIT2 levels were measured in culture supernatant using ELISA and normalized to total protein concentration in the supernatants. p=0.0204 (normoxia vs hypoxia at 12 hr), and p=0.0011 (normoxia vs hypoxia at 48 hr). Mean values ± SEM. *p<0.05, **p<0.01, and ***p<0.001. The source data are available as *Figure 5—source data 1*.

The online version of this article includes the following source data and figure supplement(s) for figure 5:

**Source data 1.** The file contains source data for *Figure 5A–D*.

**Figure supplement 1.** Dermal microvascular endothelial cells are a source of SLIT2 during *S. aureus* infection.

**Figure supplement 1—source data 1.** The file contains source data for *Figure 5—figure supplement 1A, B*.

was also observed with conditioned media from hypoxic cells (*Figure 5—figure supplement 1B*). Our findings suggest that *S. aureus* infection and local tissue oxygen levels could collectively regulate SLIT2 production by DMEC in SSTI.

## Discussion

We demonstrate here that N-SLIT2 augments extracellular ROS production by primary human and murine neutrophils in response to secondary stimulation with *S. aureus* and PMA. This effect is neutralized by incubating N-SLIT2 with N-ROBO1 and is, therefore, mediated via the canonical SLIT2-ROBO1 signaling pathway. These findings are in keeping with those of Wu et al., who reported that SLIT2 potentiates fMLP-induced oxidative burst in neutrophil-like HL-60 cells, without activating it on its own (*Wu et al., 2001*). Our work suggests that the inability of N-SLIT2 alone to induce ROS production

by neutrophils could be explained by the lack of Rac activation (*Figure 1—figure supplement 1G*), as the latter is essential for activation of the NOX complex (*Diebold and Bokoch, 2001*). We recently reported that SLIT2-ROBO1 signaling inhibits macropinocytosis in macrophages in vitro as well as in vivo (*Bhosle et al., 2020*). We show here that N-SLIT2 has no effect on phagocytosis of *S. aureus* by neutrophils and macrophages, in keeping with our previous observation that N-SLIT2 does not affect FcγR-mediated phagocytosis of opsonized particles by neutrophils (*Chaturvedi et al., 2013*), and an independent recent report showing that SLIT2 does not affect phagocytosis of *M. tuberculosis* by innate immune cells (*Borbora et al., 2023*). Together, these results suggest that N-SLIT2 can selectively promote some phagocyte functions while attenuating or not affecting others.

The phagocyte oxidative burst is a powerful tool for host defense against versatile pathogens such as *S. aureus* (*Buvelot et al., 2017*; *Neehus et al., 2021*). The oxidative burst can be further enhanced by priming agents, including tumor necrosis factor-α, which increase ROS production in response to secondary activating stimuli but have no effect on their own (*Dang et al., 2006*). In the resting state, three out of the five core components of the NOX complex, namely NCF1, NCF2, and NCF4, reside in the cytosol while the other two, CYBA and CYBB, are transmembrane proteins (*El-Benna et al., 2008*). We selectively investigated NCF1 (also known as p47phox) because of its well-characterized role in promoting NOX activation at the plasma membrane, and therefore, extracellular ROS production (*Li et al., 2010*; *Warnatsch et al., 2017*). The primed state of the NOX complex is characterized by phosphorylation of its cytosolic components, including NCF1, followed by their membrane translocation (*El-Benna et al., 2008*). We found that, upon exposure to N-SLIT2, NCF1 is phosphorylated at the conserved Serine[345] residue, which in turn results in NOX priming (*Dang et al., 2006*) in phagocytes. To date, more than a dozen cellular protein kinases, including PKC and p38 MAPKs, have been implicated in NCF1 phosphorylation (*El-Benna et al., 2009*). Terzi et al., postulated that neuronal SLIT2-ROBO2 signaling could activate PKC (*Terzi et al., 2021*). We observed that in neutrophils, N-SLIT2 did not activate PKC. These findings could be explained by the observation that in a cell-free system, the PKC-interacting protein AKAP79 binds to the C-termini of ROBO2/3, but fails to bind to the C-terminus of ROBO1 (*Samelson et al., 2015*). Our studies revealed that N-SLIT2 induced p38 activation in human neutrophils and that inhibition of p38 signaling potently attenuated the observed N-SLIT2-mediated increase in oxidative burst. These results are in accordance with the observation that SLIT2 and its orthologues can stimulate p38 MAPK signaling in non-mammalian neurons (*Campbell and Okamoto, 2013*; *Piper et al., 2006*). Our results are also in agreement with a recent study reporting hyperactivation of p38 MAPK in mice genetically overexpressing SLIT2 (*Wang et al., 2020*). Interestingly, during the oxidative burst, PKC and p38 MAPK target distinct serine residues in the NCF1 C-terminus for phosphorylation (*Dang et al., 2006*; *Fontayne et al., 2002*; *Meijles et al., 2014*). This might explain how N-SLIT2 (which activates p38) and PMA (which activates PKC) act synergistically during ROS production by human neutrophils. Together, these findings show that N-SLIT2-ROBO1 signaling primes the phagocyte NOX complex via p38 MAPK-mediated phosphorylation of NCF1. Intriguingly, SLIT2-induced p38 MAPK activation is not limited to innate immune cells. Li et al. recently demonstrated that N-SLIT2-ROBO1 signaling similarly activates p38 in pancreatic ductal adenocarcinoma cells as well as metastatic tumors but the underlying mechanism was not elucidated (*Li et al., 2023b*). Using a specific pharmacological inhibitor to disrupt cellular ROCK signaling (Y-27632), we show here that SLIT2-induced modulation of Rho/ROCK signaling is vital for its effect on p38 MAPK activity. Finally, we investigated SLIT2-induced cytoskeletal changes in neutrophils in whole blood using an established biophysical approach, RT-DC (*Bashant et al., 2019*; *Toepfner et al., 2018*). There was a trend for neutrophils from N-SLIT2-treated blood samples to be smaller and less deformed as compared to counterparts treated with N-SLIT2ΔD2, but this did not reach statistical significance (*Figure 3—figure supplement 1A–B*). This could be due to the dilution of the sample with RT-DC buffer thereby reducing the magnitude of NSLIT-2-induced effects. N-SLIT2 can also interact with other proteins present in whole blood, thereby modulating the activity of the protein.

An ever-growing body of evidence reveals phenotypic and functional heterogeneity of neutrophils in the circulation as well as in tissues (*Crainiciuc et al., 2022*; *Fine et al., 2019*; *Sagiv et al., 2015*). As some of the neutrophil subtypes are sensitive to in vitro isolation methods, a more representative snapshot of their inherent properties is enabled by optimized flow cytometry protocols that use whole blood preparations (*Fine et al., 2019*; *Fine et al., 2016*). We found that in addition to priming neutrophils, N-SLIT2 also enhanced exocytosis of secondary granules after exposure to *S. aureus*, stimulating

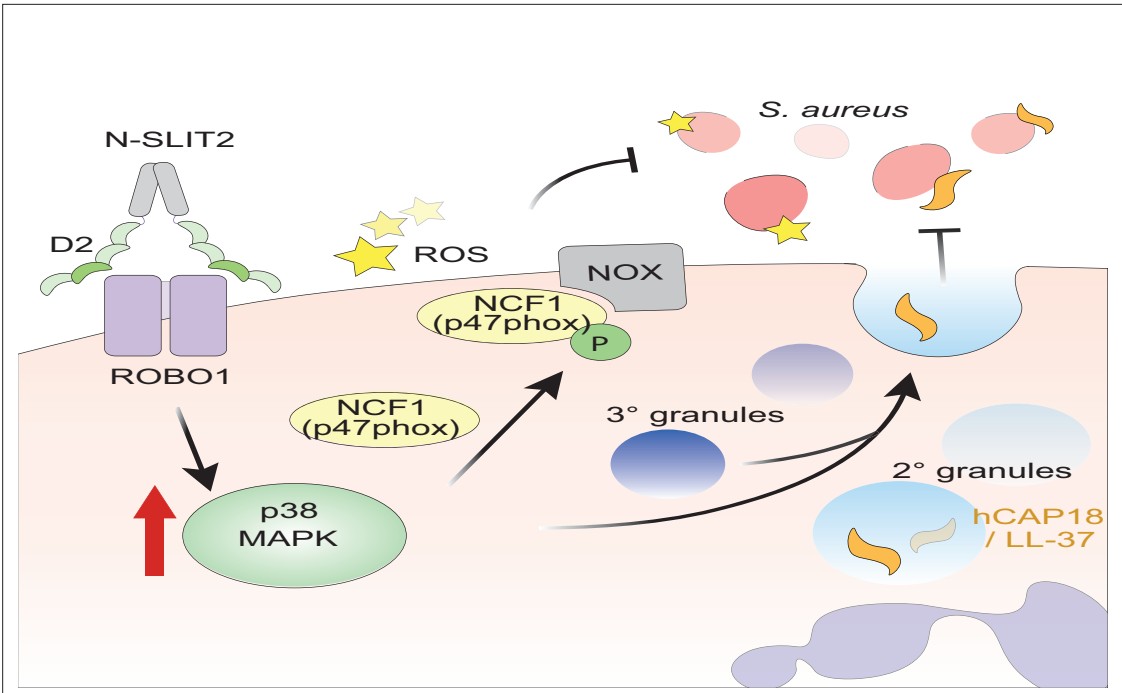

**Figure 6.** Proposed mechanism of N-SLIT2's anti-bacterial action. The binding of the Leucine-rich repeat D2 domain of N-SLIT2 to cell-surface ROBO1 results in the activation of p38 MAPK signaling in neutrophils. Active p38 MAPK phosphorylates cytosolic Neutrophil Cytosolic Factor 1 (NCF1) (p47phox), which is in its resting state, inducing translocation to the plasma membrane, thereby converting NCF1 to its primed state, forming a multi-protein NOX complex. In the presence of secondary stimuli such as *S. aureus* (and PMA), N-SLIT2-induced phosphorylation of NCF1 results in increased extracellular oxidative burst by neutrophils. The activation of p38 MAPK also augments exocytosis of secondary and tertiary granules. Secondary granules contain hCAP-18 which is cleaved extracellularly into its active form, the anti-microbial peptide, LL-37. Together, the N-SLIT2-mediated upsurge in both reactive oxygen species (ROS) production and LL-37 secretion promotes enhanced extracellular killing of *S. aureus*.

the release of LL-37, an antimicrobial cathelicidin peptide which is highly effective in killing *S. aureus* extracellularly at nanomolar concentrations (*Noore et al., 2013*). Our results are congruent with previous reports that exposure to *S. aureus* increases CD66b surface levels in neutrophils (*Mattsson et al., 2003*), but we noted that this effect was further augmented by concurrent exposure to both N-SLIT2 and *S. aureus*. This can be explained by our observation that N-SLIT2 activates p38 MAPK and primes the NOX complex. In neutrophils, priming of the NOX complex by p38 MAPK results in the exocytosis of secondary granules, and LL-37 is stored in its pro-peptide form in secondary granules (*Potera et al., 2016*; *Sørensen et al., 1997*; *Ward et al., 2000*). Indeed, we observed that inhibition of NOX partially blocked the effects of N-SLIT2, while inhibition of p38 MAPK abolished both extracellular ROS production as well as secondary granule release.

Although N-SLIT2 enhanced *S. aureus*-induced exocytosis of secondary granules in neutrophils, it did not alter cell-surface levels of the phagocytic receptor, CD16. Similarly, exposure to N-SLIT2 did not change cell-surface levels of CD63, a protein stored in primary (azurophilic) granules of neutrophils (*Borregaard and Cowland, 1997*). Our results are in line with studies showing that neither *S. aureus* nor p38 activation induces primary granule release from neutrophils (*Lu et al., 2014*; *Potera et al., 2016*). Together, these findings show that N-SLIT2-induced activation p38 MAPK is responsible for the secretion of secondary granules which contain the anti-staphylococcal peptide, LL-37 (*Figure 6*). There was a trend towards increased NETosis in neutrophils exposed to N-SLIT2 and *S. aureus* together as compared to those exposed to either *S. aureus* alone or *S. aureus* and N-SLIT2ΔD2 together, but the effect was not statistically significant. *S. aureus* is known to induce NETosis in both a NOX-dependent and –independent manner (*Douda et al., 2015*; *Parker et al., 2012*; *Pilsczek et al., 2010*). Additionally, *S. aureus* actively degrades NETs (*Thammavongsa et al., 2013*). More studies are needed to investigate the role of SLIT2-ROBO1 signaling in the modulation of NETosis.

Using a murine model of community-associated *S. aureus*-induced SSTI, we found that endogenous levels of SLIT2 protein at the site of infection were significantly reduced at 12 hr after infection,

and subsequently increased to reach a peak at 3 days post-infection. Local levels of SLIT3 were also reduced at 12 hr but did not increase more than those in uninfected controls. Congruent with our findings, a local reduction in *SLIT3* mRNA levels was recently reported in patients after *S. aureus*-induced SSTI (*Fyhrquist et al., 2019*). Importantly, we found that locally blocking endogenous SLIT2 at time points when it is most abundantly expressed resulted in enhanced *S. aureus*-induced SSTI. These results are consistent with our previous observation that the neutralization of endogenous SLIT2 enhances macrophage macropinocytosis and inflammatory cytokine production in vivo in a peritonitis model (*Bhosle et al., 2020*). We also observed a significant reduction in oxidative tissue injury, measured using 8-OHdG, in SSTI in mice treated with N-ROBO1 to neutralize endogenous SLIT2. Intriguingly, these animals also showed augmented neutrophil infiltration in SSTI which is in line with the well-established function of SLIT2 as a neutrophil chemorepellent (*Chaturvedi et al., 2013*; *Tole et al., 2009*; *Zhou et al., 2022*). Our results are also in accord with a recent report showing that *M. tuberculosis* infection upregulates SLIT2 expression in macrophages in vivo and this is associated with increased ROS signaling (*Borbora et al., 2023*). In recent years, vascular endothelial cells have emerged as an important source of endogenous SLIT2 in vivo (*Romano et al., 2018*; *Tavora et al., 2020*). It was recently reported that miR-218 is differentially secreted in milk in a bovine model of *S. aureus* mammary infection (*Ma et al., 2019*). Since SLIT2 is a well-known host gene for miR-218 in vascular endothelial cells (*Small et al., 2010*), we asked whether DMEC, an endothelial cell type involved in SSTI, can secrete SLIT2 in response to *S. aureus* infection. In accord with results from our in vivo experiments, SLIT2 expression was reduced at 12 hr and then increased at 48 hr following *S. aureus* infection of cultured DMEC cells in vitro. Exposure of HMEC-1 to hypoxia, typically found in advanced *S. aureus* SSTI-associated abscesses, also phenocopied the SLIT2 expression pattern in vitro. S. aureus infection is also known to upregulate cellular hypoxia-inducible factor 1-alpha (HIF-1α) signaling in vitro as well as in vivo (*Werth et al., 2010*; *Zhang et al., 2022*). Additionally, Li et al., very recently reported that HIF-1α negatively regulates SLIT2 expression in vascular smooth muscle cells (*Li et al., 2023a*). These findings have important implications for *S. aureus*-induced SSTI. First, *S. aureus*- and/or hypoxia-induced early upregulation of HIF-1α in DMEC could directly suppress SLIT2 production. It also opens up the possibility of multiple cell types being involved in the regulation of local levels of SLIT2 in SSTI and this will be the subject of future investigation.

Taken together, these results indicate that host-derived SLIT2 plays an important role in the spatio-temporal modulation of innate immune responses during bacterial infection. The rapid initial decrease in SLIT2 (and SLIT3) at the site of infection may serve to augment the immediate recruitment of neutrophils to combat the infection, while the rise in SLIT2 several days later may promote localized retention of recruited neutrophils and enhancement of their bactericidal properties. In this manner, spatiotemporal regulation of SLIT2/ROBO1 activity may potently direct bacterial killing and confer effective host protection against pathogenic infection. Further studies are needed to explore the precise mechanisms by which levels of SLIT2 are modulated in the face of bacterial infection and to explore the potential of SLIT2 as a novel anti-microbial therapeutic. Recent studies by independent research groups suggest that SLIT2 could not only serve as a therapeutic to combat *S. aureus*, but could have broader anti-microbial activity against a number of pathogens including *Mycobacterium tuberculosis*, intestinal pathogens, H5N1 influenza, and SARS-CoV-2 (*Borbora et al., 2023*; *Gustafson et al., 2022*; *London et al., 2010*). Given mounting concerns about the growing resistance of bacterial pathogens to conventional pharmacologic antibiotic treatment, SLIT2 may represent an attractive alternative or synergistic therapeutic strategy.

## Materials and methods

All antibodies, chemicals, reagents, and assay kits are listed in the Key Resources Table.

### Cell culture

RAW264.7 (TIB-71; RRID:CVCL_0493) cells were purchased from the American Type Culture Collection (ATCC; Manassas, VA, USA). Cells were cultured in Dulbecco's Modified Eagle's Medium (DMEM; Wisent, St-Bruno, QC, Canada) supplemented with 10% heat-inactivated fetal bovine serum (FBS; Wisent) with 1% antibiotic/antimycotic (Wisent) at 37 °C and 5% CO$_2$. FreeStyle 293-F (RRID:CVCL_D603) cells were purchased from Thermo Fisher Scientific (Mississauga, ON, Canada) and were

grown in FreeStyle 293 Expression Medium (GIBCO, Thermo Fisher Scientific). HMEC-1 (CRL-3243; RRID:CVCL_0307) cells were bought from ATCC and were maintained in EBM-2 Basal Medium (CC-3156) and EGM-2 MV Microvascular Endothelial Cell Growth Medium SingleQuots supplements (CC-4147) from Lonza (Kingston, ON, Canada) in an incubator at 37 °C and 5% $CO_2$. All cell lines were used for experiments at a passage of less than 10. All cell lines tested negative for mycoplasma using the Mycoplasma Plus PCR kit (#302008, Agilent Technologies, Santa Clara, CA, USA).

## Production of recombinant SLIT2 proteins

Large-scale production of bio-inactive human N-SLIT2ΔD2 was performed by transfecting FreeStyle 293-F cells with human N-SLIT2ΔD2 cDNA (1 ug/mL) using PEI reagent (Polyethyleneimine, linear, M.W. 25,000, Thermo Fisher Scientific) (*Patel et al., 2012*). After 5 days, the culture medium was loaded onto HisPur Ni-NTA resin (Thermo Fisher Scientific), washed with imidazole (35 mM), eluted with imidazole (250 mM), and desalted using a Pierce Zeba desalting column (7 K MWCO; Thermo Fisher Scientific). Protein molecular weight was determined by immunoblotting using anti-6XHIS-HRP conjugated antibody. Protein activity was assayed by using a spreading assay in RAW264.7 cells, as described previously (*Bhosle et al., 2020*). Recombinant human N-SLIT2 was purchased from Pepro-Tech (Cranbury, NJ, USA). Endotoxin levels in all preparations were measured using ToxinSensor Chromogenic LAL Endotoxin Assay Kit (GenScript, Piscataway, NJ, USA) and were less than 0.05 EU/ml.

## Study approvals

The protocol (#1000070040) for human blood collection from healthy adult donors was reviewed and approved by The Hospital for Sick Children (SickKids, Toronto, ON, Canada) Research Ethics Board. Written, informed consent was obtained from all participants prior to their participation. The animal studies were reviewed and approved by SickKids Animal Care Committee (ACC) (protocol #1000049735 – *S. aureus* infection) and The Centre for Phenogenomics (Toronto, ON, Canada) ACC (protocol #21–0326 – murine neutrophil isolation). The protocols for RT-DC experiments (# EK89032013, EK458102015) with human blood were approved by the ethics committee of the Technische Universität Dresden.

## Human neutrophil isolation

All reagents were filtered through Detoxi-Gel Endotoxin Removing Gel Columns (Thermo Fisher Scientific) prior to neutrophil isolation. Human neutrophils were isolated using a density-gradient separation method (*Ostrowski et al., 2020*). Briefly, 30 ml of blood was collected in sterile BD Vacutainer EDTA (BD Biosciences, Mississauga, ON, Canada) tubes and layered on the top of PolymorphPrep (Progen, Wayne, PA, USA) in a 1:1 ratio. The layered mixture was spun at 500 g (acceleration 1 and deceleration 0) for 30 min to separate polymorphonuclear neutrophils (PMN) from mononuclear cells. PMN were washed with Hank's balanced salt solution with calcium and magnesium (HBSS+/+; Wisent) and then subjected to red blood cell lysis using 0.2% NaCl followed by 1.6% NaCl for 30 s each. Neutrophils were resuspended in HBSS+/+ and used within 1 hr for all experiments.

## Murine BMDN isolation

C57BL/6 mice aged 8–12 weeks were purchased from Charles River Laboratories (St-Constant, QC, Canada). Mice were euthanized by $CO_2$ inhalation, and their tibias and femurs were resected and cut at both ends. Bone marrow cells were extracted and added to Percoll (Sigma-Aldrich, Oakville, ON, Canada) gradient solutions of 82%/65%/55% as previously described (*Tole et al., 2009*). The solution was centrifuged, and murine neutrophils were collected from an 82%/65% interface. Red blood cell lysis was performed as described above.

## *S. aureus* killing assays

*Staphylococcus aureus subsp. Aureus Rosenbach (S. aureus)* (ATCC 25923) was purchased from ATCC. The extracellular bactericidal activity of neutrophils was assayed as described previously (*Magon et al., 2020*), with some modifications. Freshly isolated neutrophils (1 × 10$^5$) were incubated with vehicle (HBSS+/+), N-SLIT2, or N-SLIT2ΔD2 for 15 min at 37 °C and then incubated with unoposonized *S. aureus* (MOI 10) for an additional 15 min at 37 °C. End-over rotation (6 rpm) was used to prevent neutrophil clumping. Tubes were centrifuged at 100 g for 5 min and neutrophil pellets were discarded.

Supernatant samples containing the extracellular bacteria were collected. Bacterial colony forming units (CFU) were determined using serial dilution followed by overnight culture in Tryptic Soy Agar (TSA) with 5% sheep blood (Remel, Lenexa, KS, USA) (*Magon et al., 2020*).

### *S. aureus* phagocytosis assays

The GFP-expressing strain *of S. aureus* USA300 LAC was grown at 37 °C overnight with shaking in Todd-Hewitt broth (BD Biosciences) with erythromycin (3 µg/ml) for selection (*Flannagan and Heinrichs, 2018*). Just before the phagocytosis assay, RAW264.7 macrophages or freshly-isolated human neutrophils were incubated with vehicle, N-SLIT2 or N-SLIT2ΔD2 for 15 min at 37 °C.

Overnight *S. aureus* USA300 LAC were diluted to $OD_{600nm}$ 0.05 in Todd-Hewitt broth containing erythromycin and cultivated until logarithmic growth ($OD_{600nm}$ 0.5) at 37 °C. Bacterial aggregates were dispersed using a 27-gauge needle (BD Biosciences) prior to infection. Unopsonized *S. aureus* was centrifuged onto RAW264.7 macrophages (MOI 10) or human neutrophils (MOI 10), and phagocytosis was allowed to proceed for 45 min at 37 °C under 5% $CO_2$. Samples were washed three times in sterile PBS and fixed in 3% paraformaldehyde (PFA) (Electron Microscopy Sciences, Hatfield, PA, USA) for 15 min at room temperature. Samples were blocked with block buffer, consisting of PBS containing 2% FBS and 2% bovine serum albumin (BSA) for 30 min at room temperature. To label bound but non-internalized bacteria, samples were incubated with human total IgG for 30 min at room temperature. Non-internalized bacteria were visualized using donkey anti-human IgG-Cy3 for 30 min at room temperature. To label the cell nuclei, samples were permeabilized with Triton-X100 (0.2%) for 10 min and incubated with DAPI for 10 min. The surface of RAW264.7 cells was labeled with Acti-stain-AF670 for 10 min at room temperature. The surface of neutrophils was labeled with Concanavalin A-AF647 for 30 min.

Confocal images were acquired using a spinning disk system (WaveFX; Quorum Technologies Inc, Puslinch, ON, Canada). The instrument consists of a microscope (Axiovert 200 M; Zeiss, Toronto, ON, Canada), scanning unit (CSU10; Yokogawa Electric Corp, Calgary, AB, Canada), electron-multiplied charge-coupled device (C9100-13; Hamamatsu Photonics), five-line (405-, 443-, 491-, 561-, and 655 nm) laser module (Spectral Applied Research, Richmond Hill, ON, Canada), and filter wheel (MAC5000; Ludl) and is operated by Volocity software version 6.3 (PerkinElmer, Waltham, MA, USA). Confocal images were acquired using a 63 x/1.4 N.A. oil objective (Zeiss) coupled with an additional 1.53 magnifying lens and the appropriate emission filter. Phagocytic Index and Phagocytic Efficiency were quantified as described below (*Westman et al., 2020*).

$$Phagocytic\ Index = \frac{Number\ of\ internalized\ S.\ aureus}{Number\ of\ phagocytes}$$

$$Phagocytic\ Efficiency = \frac{Number\ of\ internalized\ S.\ aureus}{\left(Number\ of\ internalized\ S.\ aureus\ +\ Number\ of\ bound\ S.\ aureus\right)}$$

## Extracellular ROS production

Extracellular ROS levels were measured using an isoluminol-based chemiluminescence assay (*Dahlgren et al., 2007*). Human or murine neutrophils ($5 \times 10^5$) were incubated with isoluminol (3 mmol/L) and horseradish peroxidase (6 U) in a 96-well plate and exposed to N-SLIT2 or N-SLIT2ΔD2 for 15 min immediately followed by a secondary stimulus consisting of *S. aureus* (MOI 10) or PMA (100 nM) at 37 °C. Chemiluminescence was measured every 5 min (for a total of 45 min) using a Varioskan LUX Plate Reader (Thermo Fisher Scientific).

## Rac G-LISA

Activation of Rac1/2/3 was measured by G-LISA as previously described (*Bhosle et al., 2020*). Following the experimental treatment, neutrophils were lysed using boiling (95 °C) lysis buffer with protease inhibitors (provided in the kit), and assays were performed according to the manufacturer's instructions (*Bhosle et al., 2020*). Readings were obtained at 490 nm using a Varioskan LUX Plate Reader.

## NETosis assay

NETosis assays were performed as previously described (*Douda et al., 2015*). Briefly, neutrophils were seeded at a density of $5 \times 10^4$ and exposed to N-SLIT2 or N-SLIT2ΔD2 in HBSS+/+ medium for

15 min immediately followed by a secondary stimulus of *S. aureus* (MOI 10) at 37 °C, where indicated. In parallel wells, cells were lysed using 0.5% (vol/vol) Triton X-100. The cell-impermeable nucleic acid stain, Sytox Green (5 µM) was added to all wells and the fluorescence was measured at 523 nm at 15 min intervals for 3 hr using a SpectraMax Gemini EM Fluorescence plate reader (Molecular Devices, San Jose, CA). To calculate the NETotic index, the readout from Triton X-100-treated wells was considered as 100% cell lysis (and DNA release), and the index was calculated as a percent of DNA release at each time point.

### Inhibitor treatments

Before adding N-SLIT2 to the cells, neutralization was performed by pre-incubating N-SLIT2 with N-ROBO1 at a 1:3 molar ratio for 1 hr at room temperature. In experiments using conditioned media, the media were exposed to either control IgG or N-ROBO1 at a final concentration of 1 µg/ml for 1 hr at room temperature, immediately before adding to the neutrophils. For ROCK inhibition, neutrophils were exposed to Y-27632 (10 µM) and N-SLIT2 (30 nM) simultaneously for 15 min. p38 MAPK (SB 203580 and p38 MAPK Inhibitor IV) and MEK 1/2 inhibitors (PD 184161) were used at a final concentration of 10 µM each for 15 min.

### Immunoblotting

Neutrophils were lysed in boiling RIPA lysis buffer (60 mM Tris, pH 7.5, 10% Glycerol, 1% TX-100, 150 mM NaCl, 0.5% Na-deoxycholate, 0.1% SDS, 1 mM EDTA) and 1 X protease and phosphatase inhibitor (#MSSAFE, Sigma-Aldrich). Protein lysate (30 µg) with 2 X Laemmli buffer (in 1:1 ratio) was loaded onto a 4–20% gradient gel (Bio-Rad, Mississauga, ON, Canada). The gel was transferred onto a PVDF membrane using the Trans-Blot Turbo Transfer System (Bio-Rad). Membranes were blocked for 1 hr at room temperature in a blocking buffer (5% BSA in PBS-T), with a solution containing primary antibodies. After washing and the addition of the secondary antibody, membranes were developed using SuperSignal West Dura and photographed using the Bio-Rad GelDoc system with Image Lab software. The denistometric quantifications were performed using ImageJ software, version 1.51 v (*Schindelin et al., 2012*).

### Flow cytometry

All treatments were performed with whole blood for 15 min at 37 °C. Immediately following the treatment, the samples were fixed with PFA (1.6%) for 15 min on ice. RBCs were lysed three times with Pharm Lyse (BD Biosciences) for 10 min on ice. Cells were resuspended in flow-assisted cell sorting (FACS) buffer (HBSS-/-, 1% BSA, 2 mM EDTA), labeled with an antibody cocktail for 30 min on ice in the dark, and washed three times with FACS buffer. Sample acquisition was performed using a Sony Spectral Cell Analyzer SA3800 in a standardization mode. At least $1 \times 10^5$ gated events were acquired per sample (*Fine et al., 2019*; *Fine et al., 2016*). Data were analyzed using FlowJo software (BD Biosciences, version 10). Human multicolor flow cytometry panel and gating were done as previously described (*Fine et al., 2019*).

### *S. aureus* infection of HMEC-1 cells

One day before infection, HMEC-1 cells were plated at a density of $0.3 \times 10^6$ cells per well in a six-well plate and the infection protocol was carried out as previously described (*Rollin et al., 2017*). Briefly, overnight cultures of *S. aureus* were diluted to $OD_{600nm}$ 0.05, cultivated until $OD_{600nm}$ 0.5 (logarithmic growth) at 37 °C, and resuspended in serum-free basal medium. Bacteria were added to HMEC-1 cells at MOI of 1 and incubated for 1 hr at 37 °C and 5% $CO_2$. Cells were washed three times with PBS containing 300 µg/ml gentamicin to kill extracellular bacteria and then cultured for indicated times in full medium (2 ml per well) containing 50 µg/ml gentamicin. At the end of the infection period, the conditioned medium was snap-frozen using liquid nitrogen, stored at –80 °C, and cellular RNA was isolated as described below.

### Hypoxia treatment of HMEC-1 cells

One day before specific treatments, HMEC-1 cells were plated at a density of $0.3 \times 10^6$ cells per well in two six-well plates. One plate was incubated at 37 °C in a hypoxia chamber (BioSpherix, Parish, NY) supplemented with 1% $O_2$ and 5% $CO_2$ and the other plate was concurrently incubated in a normoxic

**Table 1.** Amount of NaHCO$_3$ added to MCDB 131 medium.

| Final pH | NaHCO$_3$ stock (µl) added to 1 ml of medium | Final molarity (NaHCO$_3$) mM |
|---|---|---|
| 7.4 | 42.5 | 14 |
| 7 | 30 | 10 |
| 6.6 | 9 | 3 |

(20% O$_2$) tissue-culture incubator (37 °C and 5% CO$_2$). At the end of each culture period, conditioned media supernatants were snap-frozen using liquid nitrogen, and cellular RNA was isolated as described below.

In some experiments freshly isolated human neutrophils (10$^6$ cells per condition) were resuspended in 500 µl conditioned medium collected from HMEC-1 cells. In some cases, the conditioned medium was pre-treated with control IgG or N-ROBO1 (1 µg/ml) for 1 hr at room temperature just before resuspending the neutrophils. Neutrophils were then stimulated with *S. aureus* (MOI 10) for 15 min and the supernatant was collected by centrifuging the cells at 500 g for 5 min. The LL-37 levels in the supernatants were measured using an ELISA as described below.

## Culture of HMEC-1 cells at different extracellular pH conditions

The extracellular pH of the media was adjusted using a modified version of the protocol using different concentrations of sodium bicarbonate (NaHCO$_3$) (*Kondo and Osawa, 2017*). Endothelial Basal Medium MCDB 131 was prepared as follows: 2.9 g of MCDB 131 powder (without supplements) (#E3000-01B, United States Biological, Salem, MA) was dissolved in 250 ml of ddH$_2$O and the pH of the medium was adjusted using a stock solution of 0.33 M NaHCO$_3$ as described in *Table 1*.

The complete MCDB 131 medium was supplemented with 0.7% FBS, 10 ng/ml EGF, and 1 µg/ml hydrocortisone (all from Lonza, Kingston, ON, Canada), and filter-sterilized using a 0.22 µm membrane filter, as recommended by the manufacturer. The final pH of the complete medium was checked after incubation at 37 °C and 5% CO$_2$ for 24 hr. HMEC-1 cells were cultured in a complete MCDB 131 medium (pH 7.4) for 2 weeks before experiments. One day before the experiments, HMEC-1 cells were seeded in a six-well plate at a density of 0.3 × 10$^6$ cells per well. On the day of the experiment, the culture medium (pH 7.4) was replaced with a medium with the indicated pH (6.6, 7.0, and 7.4) for 6, 12, and 24 hr at 37 °C and 5% CO$_2$. The cells did not survive pH 6.6 beyond 24 hr. At the end of each treatment, media supernatants were snap-frozen using liquid nitrogen and RNA was isolated as described below.

## Quantitative reverse transcription-polymerase chain reaction

Total cellular RNA was isolated using the RNeasy Plus Mini Kit (#74136, Qiagen, Toronto, ON, Canada). The cDNA was synthesized from 1 µg total RNA by reverse transcription-polymerase chain reaction using the Superscript VILO MasterMix (#11755, Thermo Fisher Scientific). Finally, quantitative real-time PCR (qPCR) was performed using Power SYBR Green PCR Master Mix (#4367659, Thermo Fisher Scientific) with the following cycling parameters: 95 °C for 10 min, then 40 cycles of 95 °C for 15 s, 60 °C for 60 s. The forward and reverse primers (sequences in Key Resources Table) were used in a final concentration of 100 nM each (total reaction volume: 25 µl).

## ELISA

Following treatment of neutrophils supernatants were collected and snap-frozen in liquid nitrogen. The ELISA to measure LL-37 (#HK321, Hycult Biotech, Uden, Netherlands) was performed as per the manufacturer's protocol. In some experiments, murine ear pinnae were homogenized and the levels of murine SLIT2 and SLIT3 were measured by ELISA (SLIT2 kit, #CSB-E11039m, CUSABIO, Wuhan, China) (SLIT3 kit, #LS-F7173, LSBio, Seattle, WA, USA) as per the manufacturers' protocols (*Bhosle et al., 2020*). Human SLIT2 ELISA (for HMEC-1 conditioned media) (#CSB-E11038h, CUSABIO, Wuhan, China) was performed according to the manufacturer's instructions using a validated kit (*Jiang et al., 2022*). The optical density readings were measured on a microplate reader at 450 nm (VersaMax 190, Molecular Devices).

## Real-time deformability cytometry (RT-DC)

Citrate-anticoagulated whole blood (46 µl) was exposed to 4 µl of Vehicle (0.9% NaCl), or N-SLIT2 (30 nM) or N-SLIT2ΔD2 (30 nM) and incubated at room temperature on a shaker at 300 rpm for 45 min. Immediately after, each sample was diluted with 950 µl of RT-DC buffer (Cell Carrier, Zell-mechanik Dresden, Dresden, Germany) composed of 1 x PBS and 0.5% methylcellulose (final volume: 1 ml), and the sample viscosities were adjusted to 25 mPa s at 24 °C. RT-DC measurements were performed as described previously using an AcCellerator instrument (Zellmechanik Dresden, Dresden, Germany) (*Bashant et al., 2019*; *Otto et al., 2015*; *Toepfner et al., 2018*). RT-DC measurements were collected at a cellular flow rate of 0.12 µL/s (0.03 µL/s sample flow and 0.09 µL/s sheath flow). To exclude red blood cells (singlets and doublets), a gate for cell-size 5–16 µm parallel to and 5–20 µm perpendicular to the flow direction was applied. For acquiring measurements, an inverted microscope equipped with a high-speed EMOS camera was used and the images were captured at a frame rate of 2000 frames/s as cells reached the end of the constriction channel (*Bashant et al., 2019*; *Ekpenyong et al., 2017*). The leukocyte populations were identified based on cell size and mean brightness (*Toepfner et al., 2018*). Image analysis was carried out using ShapeOut2 software and Python 3.7 as previously described (*Bashant et al., 2019*). ShapeOut2 is available as an open-source application on GitHub (*Müller et al., 2019*). Cell deformation, a measure of how much cell shape deviates from circularity, was calculated as follows:

$$deformation = 1 - \frac{2\sqrt{\pi A}}{l}$$

where A is the convex hull area and l is the length of the convex hull perimeter.

## Epicutaneous *S. aureus* infection model

Sample sizes (*n*) for all in vivo experiments were calculated using G*Power software version 3.1.9.2 (Universität Düsseldorf, Germany) (*Faul et al., 2007*). An experimental unit (n) was defined as a single animal. All collected data points were included in the analysis (no exclusions). The epicutaneous model of murine *S. aureus* infection has been described previously (*Prabhakara et al., 2013*). C57BL/6 mice aged 8–12 weeks, of either sex, were anesthetized using an intraperitoneal injection of ketamine (50 mg/kg) and xylazine (5 mg/kg). The left ear was cleaned with 70% ethanol and pricked 10 times with an *S. aureus*-coated or sterile uncoated (mock infection) 25 G BD SafetyGlide needle (#305901, BD Biosciences). The mean inoculation dose of *S. aureus* was determined to be 5 × 10⁶ CFU/lesion, as determined by plating serial dilutions of homogenates from ear pinnae 4 hr after the infection. Mice were monitored twice daily. In some experiments, under anesthesia, mice received subcutaneous ear injections of N-ROBO1 or control IgG (7 µg) on days 2 and 3 after inoculation with *S. aureus*. Animals were euthanized using $CO_2$ inhalation on day 4 after infection; ear pinnae were cleaned, resected in 500 µl of sterile PBS, and homogenized using a Fisherbrand 150 Handheld Homogenizer (#15-340-167, Thermo Fisher Scientific). Bacterial CFUs were determined by serial dilution.

## Histopathology

Coronal sections (5 µm thick; three sections per sample) were embedded in Paraffin. Two sections per sample were counterstained with hematoxylin and eosin (H&E) at The Centre for Phenogenomics (TCP, Toronto, ON, Canada) and were scanned using 3DHistech Slide Scanner (The Hospital for Sick Children Imaging Facility, Toronto, ON, Canada) at 40 x magnification and examined under blinded conditions on a scale of 1–5 at TCP Pathology Core as defined previously (*Prabhakara et al., 2013*). The scoring system is indicated by:

0, no evidence of inflammation
1, minimal to mild focal dermal inflammatory infiltrates with or without dermal acanthosis;
2, mild multifocal dermal infiltrates with or without intraepidermal microabscesses;
3, multifocal to diffuse moderate dermal inflammatory infiltrates extending to the skeletal muscles with or without intraepidermal microabscesses;
4, marked dermal inflammatory infiltrates extending to the skeletal muscle and cartilage with or without intraepidermal microabscesses;
5, full thickness auricular necrosis or massive inflammation with myositis and chondritis.

Immunohistochemical (IHC) labeling and analyses were performed at TCP Pathology Core in a blinded manner. To examine oxidative damage and neutrophil infiltration in tissue, 8-OHdG staining and Ly6G-F4/80 labeling, respectively, were performed as previously described (*Sima et al., 2016*). Briefly, tissue samples were fixed using 10% formalin for 24 hr, and then transferred to PBS. Two 5 µm-thick paraffin-embedded coronal sections per sample were used- one for 8-OHdG labeling and one for Ly6G-F4/80 labeling. For 8-OHdG labeling, samples were immunostained in a Leica BOND-Max instrument (Leica Biosystems, Concord, ON, Canada) using the BOND Epitope Retrieval Solution 1 (#AR9961, Leica Biosystems), and incubated with a primary rabbit anti–8-OHdG antibody (#bs-1278R, Bioss, Atlanta, GA) for 1 hr (dilution 1:500 i.e. 2 µg/ml), followed by detection with a Bond Polymer Refine Detection kit (#DS9800, Leica Biosystems), containing a peroxide block, a polymer reagent, DAB (3,3'-Diaminobenzidine) chromogen, and a hematoxylin counterstain, as per the manufacturer's guidelines. Slides were scanned at 20 x using an Olympus VS-120 scanner (Evident, Tokyo, Japan). Image analysis was performed using HALO Image Analysis Platform V3.6.4134 (Indica Labs, Albuquerque, NM). For quantification, the Area Quantification Module V2.4.2 (Indica Labs) was used. The fold changes in % DAB +ve nuclei (compared to samples from mock-infected tissue) were calculated.

To label neutrophils (Ly6G$^+$F4/80$^-$) in tissues (*Chadwick et al., 2021*), formalin-fixed 5 µm coronal sections were submitted to heat-induced epitope retrieval with citrate buffer, pH 6.0 for 10 min in a pressure cooker, followed by washing with 1 x TBS containing 0.1% Tween 20 (TBS-T; EMD Millipore) and permeabilization with 0.5% Triton X-100 (BioShop, Burlington, ON, Canada) for 5 min at room temperature. The tissue sections were washed with TBS-T and then blocked in DAKO Protein Block, Serum-Free (#X0909, Agilent Technologies, Mississauga, ON, Canada) for 10 min at room temperature. The slides were incubated overnight with the following primary antibodies: Rabbit monoclonal [EPR22909-135] directed to Ly6G (1:500 i.e. 1.4 µg/ml, #ab238132, Abcam, Toronto, ON, Canada) and Rat monoclonal [CI: A3-1] directed to F4/80 (1:100 i.e. 10 µg/ml, #ab6640, Abcam). The following day, slides were washed 3 x with TBS-T, and then incubated with respective secondary antibodies: anti-rabbit IgG tagged with AF-555 (Red, #A-21428, Thermo Fisher Scientific) and anti-rat IgG tagged with AF-488 (Green, #A-11006, Thermo Fisher Scientific), for 1 hr at room temperature (both dilutions - 1:200 i.e. 10 µg/ml). The slides were washed three times with TBS-T and incubated with DAPI (Thermo Fisher Scientific) for 5 min. Finally, after washing three times with TBS-T, slides were mounted with Vectashield Vibrance Antifade mounting medium (#H-1700, Vector Labs, Newark, CA) and scanned 20 x using an Olympus VS-120 slide scanner (Evident, Tokyo, Japan). Image analysis was performed using the HALO Image Analysis Platform V3.6.4134 (Indica Labs). For the quantification of Ly6G$^+$F4/80$^-$ cells, the Highplex FL V4.2.14 module (Indica Labs) was used. The percent neutrophils (Ly6G$^+$F4/80$^-$ cells) per unit tissue area was calculated.

## Statistics

GraphPad Prism 9 was used for all statistical analyses (San Diego, CA, USA). All experiments were performed using at least 3 biological replicates; exact numbers of replicates are noted in the figure legends. For experiments using cell lines (RAW264.7, HMEC-1), a biological replicate was defined as cells from different frozen stocks and different passage numbers. For primary cells (human and murine neutrophils), each replicate was performed using cells from a different donor. For in vivo experiments, each animal was considered as a single replicate. All data are presented as mean +/- standard error of mean (SEM). Shapiro-Wilk test was first used to confirm a normal (Gaussian) distribution. For normal distribution, one-way ANOVA followed by Tukey's post hoc analysis was used. Data with non-Gaussian distribution were analyzed using a Mann-Whitney test. Statistical tests are described in the figure legends. A *p*-value of less than 0.05 was considered statistically significant. The *p*-values are reported up to four decimal places.

## Data availibility

All data supporting the findings of this study are available within the paper, its supplementary information files, and source data are provided as a Source Data file linked to this article. This study did not generate any new code.

## Biological materials

Unique biological materials used in this study will be made available upon reasonable request to the corresponding author (LAR).

# Acknowledgements

We thank Dr. Ronald S Flannagan (University of Western Ontario, London, ON, Canada) for a gift of *S. aureus* (USA300 LAC) strain expressing GFP (*Flannagan and Heinrichs, 2018*). We thank Sick-Kids Imaging Facility for the use of slide scanners and plate readers. We thank Milan Ganguly, Vivian Bradaschia, and Kyle Roberton in the Pathology Core at The Centre for Phenogenomics (Toronto, ON, Canada) for the IHC and image analysis services. We thank Uta Falke for technical assistance with the RT-DC experiments. We thank Dr. Spencer A Freeman (The Hospital for Sick Children, Toronto, ON, Canada) for helpful discussions. This work was supported by a grant from The Canadian Institutes of Health Research (CIHR) grant to LAR (PJT-169167). LAR also holds a Canada Research Chair in Vascular Inflammation and Kidney Injury. The work in the MG lab was funded by charitable donations to his lab. JW was supported by the Swedish Society of Medicine, the Foundation Blanceflor Boncompagni Ludovisi, née Bildt, and Restracomp. DAA was supported by Restracomp at The Hospital for Sick Children and an NSERC CGS-D scholarship. Work in the WLL laboratory was supported by an operating grant from the CIHR (OV2 170656); WLL also holds a Canada Research Chair in Mechanisms of Endothelial Permeability; TWWH was supported by an Ontario Graduate Scholarship award. JTM would like to thank the Canada Foundation for Innovation for infrastructure funding. FML is a holder of the Mitochondrial Innovation Initiative (Mito2i) as a scholarship. NT was supported by the Department of Pediatrics, University Hospital Carl Gustav Carus, Technische Universität Dresden, Dresden, Germany.

# Additional information

### Funding

| Funder | Grant reference number | Author |
| --- | --- | --- |
| Canadian Institutes of Health Research | PJT-169167 | Lisa A Robinson |
| Canadian Institutes of Health Research | OV2 170656 | Warren L Lee |
| Canada Foundation for Innovation | | Jason T Maynes |
| Technische Universität Dresden | Department of Pediatrics, University Hospital Carl Gustav Carus | Nicole Toepfner |
| Canada Research Chairs | Vascular Inflammation and Kidney Injury | Lisa A Robinson |
| Foundation Blanceflor Boncompagni Ludovisi, née Bildt | | Johannes Westman |
| Hospital for Sick Children Research Training Centre | RESTRACOMP | Johannes Westman Dustin A Ammendolia |
| Swedish Society of Medicine | | Johannes Westman |
| Canada Research Chairs | Mechanisms of Endothelial Permeability | Warren L Lee |
| Government of Ontario | Ontario Graduate Scholarship | Tse Wing Winnie Ho |
| Canada Foundation for Innovation | | Jason T Maynes |

| Funder | Grant reference number | Author |
|---|---|---|
| Mitochondrial Innovation Initiative | | Fatemeh Mirshafiei Langari |

The funders had no role in study design, data collection and interpretation, or the decision to submit the work for publication.

## Author contributions

Vikrant K Bhosle, Conceptualization, Investigation, Visualization, Methodology, Writing – original draft, Writing – review and editing; Chunxiang Sun, Sajedabanu Patel, Tse Wing Winnie Ho, Fatemeh Mirshafiei Langari, Zhubing Li, Manraj Sharma, Judah Glogauer, Investigation; Johannes Westman, Noah Fine, Investigation, Methodology; Dustin A Ammendolia, Visualization; Nicole Toepfner, Formal analysis, Investigation, Methodology; Mariana I Capurro, Resources, Methodology; Nicola L Jones, Michael Glogauer, Resources; Jason T Maynes, Warren L Lee, Conceptualization, Resources, Methodology; Sergio Grinstein, Conceptualization, Resources, Writing – review and editing; Lisa A Robinson, Conceptualization, Supervision, Funding acquisition, Writing – review and editing

## Author ORCIDs

Vikrant K Bhosle ⓘ https://orcid.org/0000-0002-2103-6454
Tse Wing Winnie Ho ⓘ https://orcid.org/0000-0003-4465-7859
Jason T Maynes ⓘ https://orcid.org/0000-0002-2050-3620
Sergio Grinstein ⓘ http://orcid.org/0000-0002-0795-4160
Lisa A Robinson ⓘ https://orcid.org/0000-0003-1714-1929

## Ethics

The protocol (#1000070040) for human blood collection from healthy adult donors was reviewed and approved by The Hospital for Sick Children (SickKids, Toronto, ON, Canada) Research Ethics Board. Written, informed consent was obtained from all participants prior to their participation.
The animal studies were reviewed and approved by SickKids Animal Care Committee (ACC) (protocol #1000049735 - S. aureus infection) and The Centre for Phenogenomics (Toronto, ON, Canada) ACC (protocol #21-0326 - murine neutrophil isolation).

## Decision letter and Author response

Decision letter https://doi.org/10.7554/eLife.87392.sa1
Author response https://doi.org/10.7554/eLife.87392.sa2

# Additional files

## Supplementary files

• MDAR checklist

## Data availability

All data generated or analysed during this study are included in the manuscript and supporting file.

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

# Appendix 1

## Appendix 1—key resources table

| Reagent type (species) or resource | Designation | Source or reference | Identifiers | Additional information |
|---|---|---|---|---|
| Antibody | Anti-β-Actin (Mouse monoclonal) Clone AC-15 | Sigma-Aldrich, Oakville, ON, Canada | Cat# A5441 RRID:AB_476744 | Western blot (WB)- 1:2000 1 hr @ Room temperature |
| Antibody | Anti-phospho-p38 MAPK (Thr180/Tyr182) (Rabbit polyclonal) | Cell Signaling Technology, Danvers, MA, USA | Cat# 9211 RRID:AB_331641 | WB- 1:2000 1 hr @ Room temperature |
| Antibody | Anti-Total p38 MAPK (Rabbit polyclonal) | Cell Signaling Technology, Danvers, MA, USA | Cat# 9212 RRID:AB_330713 | WB- 1:2000 Overnight (O/N) @ 4 °C |
| Antibody | Anti-phospho-p47phox (NCF1) (Ser345) (Rabbit polyclonal) | Thermo Fisher Scientific, Mississauga, ON, Canada | Cat# PA5-37806 RRID:AB_2554414 | WB- 1:1000 O/N @ 4 °C |
| Antibody | Anti-p47phox (NCF1) (Rabbit monoclonal) Clone G.207.2 | Thermo Fisher Scientific, Mississauga, ON, Canada | Cat# MA5-14778 RRID:AB_10989232 | WB- 1:1000 O/N @ 4 °C |
| Antibody | Anti-phospho-PKC Pan (Thr497) (Rabbit polyclonal) | Thermo Fisher Scientific, Mississauga, ON, Canada | Cat# PA5-38418 RRID:AB_2555019 | WB- 1:1000 1 hr @ Room temperature |
| Antibody | Anti-phospho-PKCδ (Tyr311) (Rabbit polyclonal) | Cell Signaling Technology, Danvers, MA, USA | Cat# 2055 RRID:AB_330876 | WB- 1:2000 1 hr @ Room temperature |
| Antibody | Anti-PKCδ (Rabbit polyclonal) | Cell Signaling Technology, Danvers, MA, USA | Cat# 2058 RRID:AB_10694655 | WB- 1:2000 O/N @ 4 °C |
| Antibody | Anti-His tag (Mouse monoclonal) Clone AD1.1.10 HRP-conjugated | R&D Systems, Inc. Minneapolis, MN, USA | Cat# MAB050H RRID:AB_357354 | WB- 1:2000 1 hr @ Room temperature |
| Antibody | Anti-human IgG-Cy3 (Donkey polyclonal) | Jackson ImmunoResearch Labs, West Grove, PA, USA | Cat# 709-165-149 RRID:AB_2340535 | Phagocytosis- 1:1000 in block buffer, 30 min @ Room temperature |
| Antibody | Anti-8-OHdG (Rabbit polyclonal) | Bioss Antibodies, Woburn, MA, USA | Cat# bs-1278R RRID:AB_10856120 | IHC- 1:500 for 1 hr @ Room temperature |
| Antibody | Anti-mouse Ly6g (Rabbit monoclonal) Clone: EPR22909-135 | Abcam Inc, Toronto, ON, Canada | Cat# ab238132 RRID:AB_2923218 | IHC- 1:500 O/N @ 4 °C |
| Antibody | Anti-mouse F4/80 (Rat monoclonal) Clone: Cl:A3-1 | Abcam Inc, Toronto, ON, Canada | Cat# ab6640 RRID:AB_1140040 | IHC- 1:100 O/N @ 4 °C |
| Antibody | Anti-Mouse IgG (H+L) (Goat polyclonal) Peroxidase-conjugated AffiniPure | Jackson ImmunoResearch Labs, West Grove, PA, USA | Cat# 115-035-003 RRID:AB_10015289 | WB- 1:10000 1 hr @ Room temperature |
| Antibody | Anti-Rabbit IgG (H+L) (Goat polyclonal) Peroxidase-conjugated AffiniPure | Jackson ImmunoResearch, West Grove, PA, USA | Cat# 111-035-144 RRID:AB_2307391 | WB- 1:10000 1 hr @ Room temperature |
| Antibody | Anti-Rabbit IgG (H+L) (Goat polyclonal) Cross-Adsorbed 2° Antibody, Alexa Fluor (AF-)555 | Thermo Fisher Scientific, Mississauga, ON, Canada | Cat# A-21428 RRID:AB_141784 | IHC- 1:200 1 hr @ Room temperature |

*Appendix 1 Continued on next page*

*Appendix 1 Continued*

| Reagent type (species) or resource | Designation | Source or reference | Identifiers | Additional information |
|---|---|---|---|---|
| Antibody | Anti-Rat IgG (H+L) (Goat polyclonal) Cross-Adsorbed 2° Antibody, Alexa Fluor 488 | Thermo Fisher Scientific, Mississauga, ON, Canada | Cat# A-11006 RRID:AB_141373 | IHC- 1:200 1 hr @ Room temperature |
| Antibody | PE Anti-Human CD16 (Mouse monoclonal) Clone 3G8 | BioLegend, San Diego, CA, USA | Cat# 980102 RRID:AB_2616616 | FC- 1 µl per 50 µl final volume, 30 min @ 4 °C in the dark |
| Antibody | APC/Cyanine7 Anti-Human CD11b (Mouse monoclonal) Clone ICRF44 | BioLegend, San Diego, CA, USA | Cat# 301342 RRID:AB_2563395 | FC- 1.25 µl per 50 µl final volume, 30 min @ 4 °C in the dark |
| Antibody | BV421 Anti-Human CD18 (Mouse monoclonal) Clone 6.7 | BD Biosciences, Mississauga, ON, Canada | Cat# 743370 RRID:AB_2871511 | FC- 1.25 µl per 50 µl final volume, 30 min @ 4 °C in the dark |
| Antibody | PerCP/Cyanine5.5 Anti-Human CD63 (Mouse monoclonal) Clone H5C6 | BioLegend, San Diego, CA, USA | Cat# 353020 RRID:AB_2561685 | FC- 1.25 µl per 50 µl final volume, 30 min @ 4 °C in the dark |
| Antibody | Pacific Blue Anti-Human CD14 (Mouse monoclonal) Clone HCD14 | BioLegend, San Diego, CA, USA | Cat# 325616 RRID:AB_830689 | FC- 2.5 µl per 50 µl final volume, 30 min @ 4 °C in the dark |
| Antibody | APC anti-human CD66b, eBioscience (Mouse monoclonal) Clone G10F5 | Thermo Fisher Scientific, Mississauga, ON, Canada | Cat# 17-0666-42 RRID:AB_2573152 | FC- 1.25 µl per 50 µl final volume, 30 min @ 4 °C in the dark |
| Antibody | Normal Mouse IgG (Mouse polyclonal) | Sigma-Aldrich, Oakville, ON, Canada | Cat# 12–371 RRID:AB_145840 | Flow cytometry (FC)- 2 µg, block 20 min @ 4 °C |
| Antibody | Human IgG control (Human polyclonal) | Sigma-Aldrich, Oakville, ON, Canada | Cat# I4506 RRID:AB_1163606 | Phagocytosis- 1:1000 in block buffer, 30 min @ Room temperature |
| Antibody | InVivoMAb human IgG$_1$ (Human polyclonal) Isotype Control | Bio X Cell, Lebanon, NH, USA | Cat# BE0297 RRID:AB_2687817 | In vivo- 7 µg per injection per mouse |
| Sequence-based reagent | SLIT2_F | This paper | PCR primers | TCCTCCTCGCACCTTTGATGGATT |
| Sequence-based reagent | SLIT2_R | This paper | PCR primers | AGAGGGTTGGCTCCAATTGCTAGA |
| Sequence-based reagent | GAPDH_F | This paper | PCR primers | GGTGTGAACCATGAGAAGTATGA |
| Sequence-based reagent | GAPDH_R | This paper | PCR primers | GAGTCCTTCCACGATACCAAAG |
| Chemical compound, drug | Acti-stain-AF670 | Universal Biologicals, Cambridge, UK | Cat# PHDN1-A | Phagocytosis |
| Commercial assay or kit | BOND Epitope Retrieval Solution 1 | Leica Biosystems, Concord, ON, Canada | Cat# AR9961 | IHC Antigen Retrieval |
| Chemical compound, drug | Concanavalin A-AF647 | Thermo Fisher Scientific, Mississauga, ON, Canada | Cat# C21421 | Phagocytosis |
| Chemical compound, drug | Ethylenediaminetetraacetic acid (EDTA), 0.5 M, pH 8.0, Sterile | Bio-World, Dublin, OH, USA | Cat# 40520000 CAS# 60-00-4 | Flow Cytometry buffer ingredient |

*Appendix 1 Continued on next page*

Appendix 1 Continued

| Reagent type (species) or resource | Designation | Source or reference | Identifiers | Additional information |
|---|---|---|---|---|
| Commercial assay or kit | Detoxi-Gel Endotoxin Removing Gel | Thermo Fisher Scientific, Mississauga, ON, Canada | Cat# 20339 | Endotoxin removal |
| Commercial assay or kit | Detoxi-Gel Endotoxin Removing Gel Columns | Thermo Fisher Scientific, Mississauga, ON, Canada | Cat# 20344 | Endotoxin removal |
| Chemical compound, drug | Gentamicin (10 mg/mL) | Thermo Fisher Scientific, Mississauga, ON, Canada | Cat# 15710064 | HMEC-1 culture to selectively kill extracellular *S. aureus* bacteria |
| Chemical compound, drug | Isoluminol (4-Aminophthalhydrazide) | Sigma-Aldrich, Oakville, ON, Canada | Cat# A8264 CAS# 3682-14-2 | Extracellular ROS measurement |
| Chemical compound, drug | p38 MAPK Inhibitor IV | Cayman Chemical, Ann Arbor, MI, USA | Cat# 22219 CAS# 1638-41-1 | p38 MAPK inhibitor |
| Chemical compound, drug | Paraformaldehyde 16% solution | Electron Microscopy Sciences, Hatfield, PA, USA | Cat# 15710 CAS# 50-00-0 | Fixative |
| Chemical compound, drug | PD 184161 | Cayman Chemical, Ann Arbor, MI, USA | Cat# 10012431 CAS# 212631-67-9 | MEK1/2 inhibitor |
| Commercial assay or kit | Percoll | Sigma-Aldrich, Oakville, ON, Canada | Cat# P1644 | Murine neutrophil isolation |
| Commercial assay or kit | PolymorphPrep | Progen, Wayne, PA, USA | Cat# 1895 | Human neutrophil isolation |
| Chemical compound, drug | Power SYBR Green PCR Master Mix | Thermo Fisher Scientific, Mississauga, ON, Canada | Cat# 4367659 | Quantitative PCR |
| Chemical compound, drug | Phorbol 12-myristate 13-acetate (PMA) | Sigma-Aldrich, Oakville, ON, Canada | Cat# P8139 CAS #16561-29-8 | Neutrophil activation |
| Peptide, recombinant protein | Recombinant Human N-SLIT2 | PeproTech, Cranbury, NJ, USA | Cat# 150–11 | Neutrophil treatments |
| Peptide, recombinant protein | Recombinant Human ROBO1 Fc Chimera (N-ROBO1) | R&D Systems, Minneapolis, MN, USA | Cat# 8975-RB | Neutrophil treatment |
| Chemical compound, drug | SB 203580 | Sigma-Aldrich, Oakville, ON, Canada | Cat# S8307 CAS #152121-47-6 | p38 MAPK inhibitor |
| Commercial assay or kit | Superscript VILO MasterMix | Thermo Fisher Scientific, Mississauga, ON, Canada | Cat# 11755 | Reverse Transcription |
| Commercial assay or kit | SYTOX Green Nucleic Acid Stain | Thermo Fisher Scientific, Mississauga, ON, Canada | Cat# S7020 | NETosis assay |
| Chemical compound, drug | Y-27632, ROCK inhibitor | Sigma-Aldrich, Oakville, Canada | Cat# SCM075 CAS# 331752-47-7 | Neutrophil treatment |

Appendix 1 Continued on next page

*Appendix 1 Continued*

| Reagent type (species) or resource | Designation | Source or reference | Identifiers | Additional information |
|---|---|---|---|---|
| Commercial assay or kit | Bond Polymer Refine Detection kit | Leica Biosystems, Concord, ON, Canada | Cat# DS9800 | 8-OHdG (DAB) staining |
| Commercial assay or kit | G-LISA Rac Activation Assay | Cytoskeleton, Inc, Denver, CO, USA | Cat# BK125 | Rac1/2/3 activation assay |
| Commercial assay or kit | Human LL-37 ELISA kit | Hycult Biotech, Uden, Netherlands | Cat# HK321 | ELISA |
| Commercial assay or kit | Human SLIT2 ELISA kit | Cusabio, Wuhan, P.R. China | Cat# CSB-E11038h | ELISA |
| Commercial assay or kit | Mouse SLIT2 ELISA Kit | Cusabio, Wuhan, P.R. China | Cat# CSB-E11039m | ELISA |
| Commercial assay or kit | Mouse SLIT3 ELISA kit | Lifespan Biosciences Seattle, WA, USA | Cat# LS-F7173 | ELISA |
| Commercial assay or kit | RNeasy Plus Mini Kit | Qiagen, Toronto, ON, Canada | Cat# 74136 | RNA isolation |
| Cell line (*H. sapiens*) | FreeStyle 293-F Cells (HEK293F) | Thermo Fisher Scientific, Mississauga, ON, Canada | R79007 RRID:CVCL_D603 | N-SLIT2ΔD2 production |
| Cell line (*H. sapiens*) | HMEC-1 | American Type Culture Collection (ATCC), Manassas, VA, USA | CRL-3243 RRID:CVCL_0307 | Immortalized human dermal microvascular endothelial cells |
| Cell line (*M. musculus*) | RAW264.7 | American Type Culture Collection (ATCC), Manassas, VA, USA | TIB-71 RRID: CVCL_0493 | Murine macrophage cell line |
| Strain, strain background (*Staphylococcus aureus*) | *Staphylococcus aureus* GFP USA300 LAC strain | Dr. Ronald S. Flannagan (University of Western Ontario, London, ON, Canada) PMID: 30619165 | *Staphylococcus aureus* GFP | Phagocytosis |
| Strain, strain background (*Staphylococcus aureus*) | *Staphylococcus aureus* subsp. Aureus Rosenbach | American Type Culture Collection (ATCC), Manassas, VA, USA | ATCC 25923 | *S. aureus* (all experiments except phagocytosis) |
| Software, algorithm | ShapeOut2 | PMID: 29331015 | ShapeOut2; ***Müller et al., 2019*** | https://github.com/ZELLMECHANIK-DRESDEN/ShapeOut2 |

