## [Editor Report]

Bhosle and colleagues present valuable findings on the function of the N-terminal fragment of SLIT2 in the amplification of reactive oxygen species production and exocytosis of secretory granules by neutrophils. The authors present solid in vitro and in vivo data supporting this unexpected role for SLIT2. This work advances our knowledge of innate immunity to pathogens.

---

## [Decision Letter]

[Editors' note: this paper was reviewed by Review Commons.]

---

## [Author Response]

General Statements

We sincerely thank the reviewers for their comprehensive and constructive feedback. Below, we submit our revision plan addressing the points raised by the reviewers.

Description of the planned revisionsReviewer #1 (Evidence, reproducibility and clarity (Required)):

The study analyzes the role of SLIT2 in clearance of S. aureus via neutrophils. It suggests that N-SLIT2 play a key role as an amplifier of the ROS response and release of antimicrobial peptides. The manuscript is well written and is technologically sound. However, a few issues need to be addressed that preclude publication of the manuscript:

We thank the reviewer for the positive feedback.

Major comments:1. The study analyzes different parameters of neutrophil function. One major effect of neutrophil activation is NETosis. This has not been addressed in the study albeit it is deemed to act in concert with the other immune mechanisms described.

We thank the reviewer for the suggestion. To investigate whether N-SLIT2 modulates *S. aureus*-induced Neutrophil Extracellular Trap formation (NETosis), we used Sytox Green, a membrane-impermeable nucleic acid label and calculated the NETotic index, as previously described (Douda et al., 2015). Exposure of neutrophils to N-SLIT2 in the absence of bacteria did not induce NETosis. In line with previously published results (Monteith et al., 2021; Wan et al., 2017), incubation of neutrophils with *S. aureus* (MOI 10) for 2 hours resulted in significant NETosis (p = 0.0004). Incubation of neutrophils with N-SLIT2 and *S. aureus* showed a trend towards higher NETosis compared to incubation with *S. aureus* alone, but this did not reach statistical significance (p = 0.0883). *S. aureus* is known to induce NETosis in both NADPH oxidase (NOX)–dependent and –independent manners (Douda et al., 2015; Parker et al., 2012; Pilsczek et al., 2010). Additionally, *S. aureus* is known to promote the degradation of neutrophil-derived NETs (Thammavongsa et al., 2013). Our results demonstrate that N-SLIT2-ROBO1 signaling primes p47phox, a component of neutrophil NOX2 complex, in a p38-dependent manner but the contribution of NOX-independent NETosis as well as degradation of NETs by *S. aureus* cannot be presently ruled out. New studies are warranted to investigate the effects of SLIT-ROBO signaling on pathogen-induced NETosis. In the revised manuscript, we included the new results and discussed these important points in revised Figure 3—figure supplement 1H-J and on pages 7 and 10 of the revised manuscript.

**Author response image 1. sa2fig1:** 

2. Furthermore, the authors discuss a role of SLIT2 in the regulation of neutrophil migration. However, the current data set does not provide sufficient evidence for this. The reviewer suggests that the authors provide migration/chemotaxis assays and/or in vivo data to prove their hypothesis or revise their argumentation.

We thank the reviewer for this important suggestion. Several groups, including ours, have previously demonstrated that SLIT2-ROBO1 signaling potently inhibits neutrophil chemotaxis in vitro and in vivo (Chaturvedi et al., 2013; Tole et al., 2009; Ye et al., 2010; Zhou et al., 2022). Additionally, a recent study showed that shRNA-mediated knockdown of SLIT2 resulted in increased neutrophil infiltration into murine tumors further supporting the negative regulatory effect of SLIT2 on neutrophil migration (Geraldo et al., 2021). In our current study, in an effort to selectively examine the effects of SLIT2 on neutrophil function rather than on neutrophil migration, we intentionally administered N-ROBO1 to block endogenous SLIT2 signaling at 48 and 72 hours after inducing skin and soft tissue infection (SSTI) with *S. aureus*. In this model, the majority of neutrophil influx occurs early on, namely within 24 hours (Prabhakara et al., 2013). We observed that blocking endogenous SLIT2 signaling in a murine model of SSTI resulted in enhanced localized infection and injury. We have now performed immunohistochemistry (IHC) to evaluate within the tissue levels of 8-hydroxydeoxyguanosine (8-OHdG), an indicator of oxidative tissue damage which has previously been shown to be associated with neutrophil activity (Sima et al., 2016). Congruent with the H&E results, neutralizing endogenous SLIT2 signaling using N-ROBO1 resulted in decreased oxidative damage in SSTI (Figure 4—figure supplement 1F-G). Next, we used IHC staining to directly measure neutrophil (Ly6G^+^F4/80^-^) numbers in SSTI (Chadwick et al., 2021). In line with aforementioned studies demonstrating the potent ability of exogenously administered SLIT2 to inhibit neutrophil chemotaxis in murine models of non-infectious diseases, we found that neutralization of endogenous SLIT2, using N-ROBO1, augmented neutrophil infiltration in SSTI.

Interestingly, a small but significant increase in neutrophil infiltration was also observed in animals treated with N-ROBO1 alone in the absence of SSTI (Figure 4—figure supplement 1D-E). We now discuss these important points on pages 7 and 11 of the revised manuscript.

3. The timeline of SLIT2 expression indicates that environmental conditions could influence the expression of SLIT2. Have the authors analyzed whether SLIT2 expression is affected by low pH or hypoxia? Is there any data indicating what factors regulate SLIT2 expression? In the same line, it would be interesting to know whether SLIT2 immune effects (specifically ROS and LL37 release) are similarly triggered under hypoxic conditions often found in an abscess.

We thank the reviewer for raising this important point and for the suggestions. The regulation of SLIT2 levels in tissues is an active area of research. Hypoxia has been reported to increase SLIT2 expression in placental tissue (Liao et al., 2012), but this has not been investigated in the context of bacterial infection. In different physiologic and pathophysiologic settings, vascular endothelial cells, including dermal microvascular endothelial cells, have been shown to be an important source of SLIT2 (Romano et al., 2018; Tavora et al., 2020). We performed additional experiments using HMEC-1, a human dermal microvascular endothelial cell line (Ades et al., 1992) to investigate effects of hypoxia and low pH, conditions found within bacterial abscesses, on production of SLIT2. In line with tissue levels of SLIT2 protein in *S. aureus*-induced SSTI (Figure 4A), SLIT2 production by HMEC-1 cells was significantly attenuated at 12 hours, but then increased at 48 hours after infection with *S. aureus* both at mRNA and protein levels (Figure 5 A-B). Lowering the pH of the cell culture media to pH 7.0 and 6.6 did not significantly alter SLIT2 production by HMEC-1 cells (Figure 5—figure supplement 1A). To test the effects of hypoxia on SLIT2 production we exposed dermal microvascular endothelial cells to 1% O_2_ and observed significant reduction in SLIT2 levels at 12 hours and an increase at 48 hours (Figure 5 C-D). While our manuscript was in revision, Li *et al.* reported that cellular levels of Hypoxia-inducible factor 1-α (HIF-1α) negatively regulate SLIT2 production in vascular smooth muscle cells, with higher HIF-1α levels suppressing SLIT2 production and vice versa (Li et al., 2023b). Intriguingly, *S. aureus* infection is also known to activate HIF-1α signaling in vitro and in vivo (Thompson et al., 2017; Werth et al., 2010; Zhang et al., 2022). Taken together, these results suggest that temporal regulation of local HIF-1α signaling could regulate endogenous SLIT2 levels in SSTI. We have included the results of our new experiments and also discuss the important points above in the Discussion section of the revised manuscript (Figure 5 and page 11).

Finally, we investigated the effects of SLIT2, produced by HMEC-1 cells, on LL-37 production by treating neutrophils with conditioned media from HMEC-1 cells. The conditioned medium from *S. aureus*-infected HMEC-1 cells (48 h) stimulated LL-37 from neutrophils but this effect was blocked by treatment of conditioned medium with N-ROBO1, which contains the N-SLIT2-binding Ig1 motif (Morlot et al., 2007) and blocks the action of bioactive N-SLIT2 (p = 0.0362) (Figure 5—figure supplement 1B). Similarly, conditioned medium from HMEC-1 cells subjected to hypoxia (48 h) stimulated LL-37 release but this was significantly attenuated by N-ROBO1 treatment (p = 0.0190) (Figure 5—figure supplement 1B). We discuss these new results on page 8 of the revised manuscript.

4. Lastly, it is unclear whether SLIT2 binds to a defined target on the neutrophil. This needs to be highlighted in the discussion in respect to future work and ideally resolved experimentally.

We apologize for the confusion. We and others have previously demonstrated that human and murine neutrophils express ROBO1 but not ROBO2, and that ROBO1 is the primary Roundabout receptor which binds N-SLIT2 in neutrophils (Rincon et al., 2018; Tole et al., 2009). We have now included this information in the Introduction section (please see page 4 of the revised manuscript). In our manuscript we showed experimentally that incubation of N-SLIT2 with the soluble N-terminal fragment of ROBO1 (N-ROBO1) blocked the effect of N-SLIT2 on ROS production, thereby confirming that the observed actions of SLIT2 occurred through ROBO1 (Figure 1G). In the revised version of the manuscript, we clarify this point in the Results section (page 5).

Reviewer #1 (Significance (Required)):The manuscript provides insight into a new mechanism regulating neutrophil function in the presence of S. aureus. The study provides evidence that the N-terminus of SLIT2 is involved in these effects and highlights p38-mediated signaling events as molecular targets increasing antibacterial effects in neutrophils. However, some contradictory findings imply that timing of the response is crucial.Nevertheless, with the molecular mechanisms not fully understood many questions remain and the study adds to the complexity of the anti-staphylococcal immune response. Therefore, the audience for this manuscript requires knowledge on S. aureus-specific host-pathogen interaction and is not suitable for a broad audience as it does not provide information on a generally new mechanism of neutrophil activation or defense.

We thank the reviewer for highlighting the point that the temporal control of endogenous SLIT2 levels is crucial for modulating innate immune responses. We complete agree and clarify and expand upon this important point in the revised manuscript (page 11).

To more fully elucidate the molecular and cellular mechanisms responsible for our observations, we performed a series of new experiments. We recently reported that exposure to N-SLIT2 increases cell rounding and decreases spreading in macrophages and this effect is mediated by inactivation of the endogenous RhoGAP, MYO9B (Bhosle et al., 2020). Using novel methodology, neutrophil priming was recently shown to be associated with characteristic cytoskeletal changes (Bashant et al., 2019). We, therefore, collaborated with Dr. Nicole Toepfner (Technische Universität Dresden, Dresden) to investigate SLIT2-induced cytoskeletal changes in neutrophils isolated from whole blood using Real-time deformability cytometry (RT-DC), a technique that has recently been utilized to investigate priming-associated cytoskeletal changes in neutrophils in situ (Bashant et al., 2019; Toepfner et al., 2018). In line with aforementioned findings in macrophages, exposure to bioactive N-SLIT2, but not bio-inactive N-SLIT2ΔD2 resulted in approximately 10% reduction in average cell area for neutrophils although the difference did not reach statistical significance (vehicle vs N-SLIT2, p = 0.0676) (Figure 3—figure supplement 1B). Additionally, N-SLIT2-exposed neutrophils were less deformed as compared to the N-SLIT2ΔD2 (vehicle vs N-SLIT2, p = 0.0955) (Figure 3—figure supplement 1A). This could be due to the dilution of the sample with RT-DC buffer thereby reducing the magnitude of NSLIT-2-induced cellular effects. N-SLIT2 can also interact with other proteins present in whole blood, thereby modulating activity of the protein. We discuss these points on page 10 of the revised manuscript.

Congruent with our results in innate immune cells (neutrophils and macrophages), Li et al. very recently reported that SLIT2-ROBO1 signaling similarly stimulates p38 MAPK activity in human cancer cells, but the underlying mechanism has remained elusive so far (Li et al., 2023a). Rho-associated kinase (ROCK) signaling is known to augment p38 MAPK activity in epithelial cells (Wagstaff et al., 2016). Since we and others have shown that SLIT2-ROBO1 signaling enhances RhoA activation in immune and cancer cells (Bhosle et al., 2020; Kong et al., 2015), we tested the effect of ROCK inhibition on SLIT2-induced p38 activation in neutrophils. In the presence of the specific ROCK inhibitor, Y-27632, N-SLIT2 failed to activate p38 MAPK in neutrophils (p = 0.0007) (Figure 2G-H). We discuss these results on page 10 of the revised manuscript.

We also thank the reviewer for pointing out the complexity of host-pathogen interactions involving neutrophils and *S. aureus*. SLIT2 is well-known for its anti-inflammatory properties via its effects on immune cell chemotaxis in vivo (Anand et al., 2013; Chaturvedi et al., 2013; Geraldo et al., 2021). We demonstrated that SLIT2-ROBO1 signaling inhibits macropinocytosis in macrophages, and consequently, attenuates NOD2-induced inflammasome activation in mice (Bhosle et al., 2020). Based on these earlier observations, SLIT2 would be anticipated to impair innate immune responses to infection. Unexpectedly, we found that SLIT2 does not impair, but instead enhances the ability of neutrophils to kill *S. aureus*. Indeed, through different mechanisms SLIT2 has been shown to have widespread immunomodulatory properties against not only *S. aureus* but against diverse pathogens, including *M. tuberculosis,* intestinal pathogens*,* H5N1 influenza, and most recently, COVID-19 (Borbora et al., 2023; Gustafson et al., 2022; London et al., 2010). Together, these studies highlight the importance of spatiotemporal regulation of SLIT2 levels in tissues during bacterial and viral infections and the direct effects of SLIT2 on modulating host-pathogen interactions. In the revised manuscript, we now discuss these important points (page 11) in the revised manuscript.

Reviewer #2 (Evidence, reproducibility and clarity (Required)):Summary:The manuscript deals with the role of the neurorepellent SLIT2 in killing of the bacterial pathogen Staphylococcus aureus. The authors show that neutrophils incubated with the N-terminal region of SLIT2 kill S. aureus more efficiently than neutrophils without pre-exposure to N-SLIT2. This effect was due to an increased production of reactive oxygen species by NADPH oxidase complex activation and stimulating exocytosis of antibacterial peptide containing granules. The concept was proven in an animal model of skin and soft tissue infection in mice in which neutralization of endogenous SLIT2 reduced CFU numbers in ear skin and decreased tissue destruction in response to S. aureus infection.Major comments:1. In general the findings and key conclusions are convincingly covered by the results presented in the manuscript. The methods are adequate to allow the conclusions drawn. Data are clearly presented and easy to follow. Statistical methods are appropriate.

We thank the reviewer for the positive feedback.

Minor comments:1. In the Materials and methods section line 340 a GFP-expressing S. aureus USA300 strain is indicated. What was the exact strain designation, e.g. LAC or JE2, as USA300 is not a strain name (different strains belong to this pulsed-field electrophoresis based classification).

We thank the reviewer for this comment. The strain designation of the GFP-expressing *S. aureus* we used is USA300 LAC (Flannagan et al., 2018). In the revised version of the manuscript we have now included the correct information (please see page 13).

2. In the legend of figure 3 the inhibitors are mentioned for part B and E but not C and D.

We apologize for the error. The legend of Figure 3 has now been corrected and is now included in the revised manuscript.

3. Figure S4 would be nice to have in the main manuscript.

We thank the reviewer for the suggestion. In the revised manuscript we moved the original Supplementary Figure S4B to main Figure 4B. The schematic from main Figure 4B has been moved to the new Supplementary Figure 4B (Figure 4—figure supplement 1B). The graphical summary is now presented as the new main Figure 6.

Reviewer #2 (Significance (Required)):The manuscript deals with a novel mechanism of neutrophil activation by SLIT-2, a protein which was originally thought to act in the nervous system but is also expressed in many peripheral tissues. Importantly SLIT-2 may be involved in tumor suppression but also chemotaxis of immune cells. In this manuscript a novel, rather unexpected role of the N-terminal region of SLIT-2 in activation of antibacterial mechanisms of neutrophils was shown. This could be interesting for a broader readership interested in innate immune mechanisms and bacterial infections. Since little is known on the role of SLIT-2 in response to bacterial infections the paper could initiate a number of new studies in this field. This reviewer has experience with S. aureus virulence and resistance mechanisms and animal infection models.

We thank the reviewer for the very positive feedback regarding the appeal of our manuscript to a broad readership. As noted in our response to Reviewer #1 Significance, recent studies suggest that SLIT2 could not only serve as a therapeutic to combat *S. aureus*, but could have broad anti-microbial activity against a number of pathogens including *Mycobacterium tuberculosis,* intestinal pathogens*,* H5N1 influenza, and COVID-19 (Borbora et al., 2023; Gustafson et al., 2022; London et al., 2010). We believe that the ability of SLIT2 to combat diverse bacterial and viral infections will even further enhance the appeal of our manuscript to a broad audience. In the revised manuscript we expand the discussion (page 11) to include these very important points.

References:

Ades, E.W., Candal, F.J., Swerlick, R.A., George, V.G., Summers, S., Bosse, D.C., and Lawley, T.J. (1992). HMEC-1: establishment of an immortalized human microvascular endothelial cell line. J Invest Dermatol *99*, 683-690.

Anand, A.R., Zhao, H., Nagaraja, T., Robinson, L.A., and Ganju, R.K. (2013). N-terminal Slit2 inhibits HIV-1 replication by regulating the actin cytoskeleton. Retrovirology *10*, 2.

Bashant, K.R., Vassallo, A., Herold, C., Berner, R., Menschner, L., Subburayalu, J., Kaplan, M.J., Summers, C., Guck, J., Chilvers, E.R.*, et al.* (2019). Real-time deformability cytometry reveals sequential contraction and expansion during neutrophil priming. J Leukoc Biol *105*, 1143-1153.

Bhosle, V.K., Mukherjee, T., Huang, Y.W., Patel, S., Pang, B.W.F., Liu, G.Y., Glogauer, M., Wu, J.Y., Philpott, D.J., Grinstein, S.*, et al.* (2020). SLIT2/ROBO1-signaling inhibits macropinocytosis by opposing cortical cytoskeletal remodeling. Nat Commun *11*, 4112.

Borbora, S.M., Satish, B.A., Sundar, S., B, M., Bhatt, S., and Balaji, K.N. (2023). *Mycobacterium tuberculosis* elevates SLIT2 expression within the host and contributes to oxidative stress responses during infection. J Infect Dis.

Chadwick, J.W., Macdonald, R., Ali, A.A., Glogauer, M., and Magalhaes, M.A. (2021). TNFalpha Signaling Is Increased in Progressing Oral Potentially Malignant Disorders and Regulates Malignant Transformation in an Oral Carcinogenesis Model. Front Oncol *11*, 741013.

Chaturvedi, S., Yuen, D.A., Bajwa, A., Huang, Y.W., Sokollik, C., Huang, L., Lam, G.Y., Tole, S., Liu, G.Y., Pan, J.*, et al.* (2013). Slit2 prevents neutrophil recruitment and renal ischemia-reperfusion injury. J Am Soc Nephrol *24*, 1274-1287.

Douda, D.N., Khan, M.A., Grasemann, H., and Palaniyar, N. (2015). SK3 channel and mitochondrial ROS mediate NADPH oxidase-independent NETosis induced by calcium influx. Proc Natl Acad Sci U S A *112*, 2817-2822.

Flannagan, R.S., Kuiack, R.C., McGavin, M.J., and Heinrichs, D.E. (2018). Staphylococcus aureus Uses the GraXRS Regulatory System To Sense and Adapt to the Acidified Phagolysosome in Macrophages. mBio *9*.

Geraldo, L.H., Xu, Y., Jacob, L., Pibouin-Fragner, L., Rao, R., Maissa, N., Verreault, M., Lemaire, N., Knosp, C., Lesaffre, C.*, et al.* (2021). SLIT2/ROBO signaling in tumor-associated microglia and macrophages drives glioblastoma immunosuppression and vascular dysmorphia. J Clin Invest *131*.

Gustafson, D., Ngai, M., Wu, R., Hou, H., Schoffel, A.C., Erice, C., Mandla, S., Billia, F., Wilson, M.D., Radisic, M.*, et al.* (2022). Cardiovascular signatures of COVID-19 predict mortality and identify barrier stabilizing therapies. EBioMedicine *78*, 103982.

Kong, R., Yi, F., Wen, P., Liu, J., Chen, X., Ren, J., Li, X., Shang, Y., Nie, Y., Wu, K.*, et al.* (2015). Myo9b is a key player in SLIT/ROBO-mediated lung tumor suppression. J Clin Invest *125*, 4407-4420.

Li, Q., Zhang, X.X., Hu, L.P., Ni, B., Li, D.X., Wang, X., Jiang, S.H., Li, H., Yang, M.W., Jiang, Y.S.*, et al.* (2023a). Coadaptation fostered by the SLIT2-ROBO1 axis facilitates liver metastasis of pancreatic ductal adenocarcinoma. Nat Commun *14*, 861.

Li, S., Gao, Z., Li, H., Xu, C., Chen, B., Zha, Q., Yang, K., and Wang, W. (2023b). Hif-1alpha/Slit2 Mediates Vascular Smooth Muscle Cell Phenotypic Changes in Restenosis of Bypass Grafts. J Cardiovasc Transl Res.

Liao, W.X., Laurent, L.C., Agent, S., Hodges, J., and Chen, D.B. (2012). Human placental expression of SLIT/ROBO signaling cues: effects of preeclampsia and hypoxia. Biol Reprod *86*, 111.

London, N.R., Zhu, W., Bozza, F.A., Smith, M.C., Greif, D.M., Sorensen, L.K., Chen, L., Kaminoh, Y., Chan, A.C., Passi, S.F.*, et al.* (2010). Targeting Robo4-dependent Slit signaling to survive the cytokine storm in sepsis and influenza. Sci Transl Med *2*, 23ra19.

Monteith, A.J., Miller, J.M., Maxwell, C.N., Chazin, W.J., and Skaar, E.P. (2021). Neutrophil extracellular traps enhance macrophage killing of bacterial pathogens. Sci Adv *7*, eabj2101.

Morlot, C., Thielens, N.M., Ravelli, R.B., Hemrika, W., Romijn, R.A., Gros, P., Cusack, S., and McCarthy, A.A. (2007). Structural insights into the Slit-Robo complex. Proc Natl Acad Sci U S A *104*, 14923-14928.

Parker, H., Dragunow, M., Hampton, M.B., Kettle, A.J., and Winterbourn, C.C. (2012). Requirements for NADPH oxidase and myeloperoxidase in neutrophil extracellular trap formation differ depending on the stimulus. J Leukoc Biol *92*, 841-849.

Pilsczek, F.H., Salina, D., Poon, K.K., Fahey, C., Yipp, B.G., Sibley, C.D., Robbins, S.M., Green, F.H., Surette, M.G., Sugai, M.*, et al.* (2010). A novel mechanism of rapid nuclear neutrophil extracellular trap formation in response to Staphylococcus aureus. J Immunol *185*, 7413-7425.

Prabhakara, R., Foreman, O., De Pascalis, R., Lee, G.M., Plaut, R.D., Kim, S.Y., Stibitz, S., Elkins, K.L., and Merkel, T.J. (2013). Epicutaneous model of community-acquired Staphylococcus aureus skin infections. Infect Immun *81*, 1306-1315.

Rincon, E., Rocha-Gregg, B.L., and Collins, S.R. (2018). A map of gene expression in neutrophil-like cell lines. BMC Genomics *19*, 573.

Romano, E., Manetti, M., Rosa, I., Fioretto, B.S., Ibba-Manneschi, L., Matucci-Cerinic, M., and Guiducci, S. (2018). Slit2/Robo4 axis may contribute to endothelial cell dysfunction and angiogenesis disturbance in systemic sclerosis. Ann Rheum Dis *77*, 1665-1674.

Sima, C., Aboodi, G.M., Lakschevitz, F.S., Sun, C., Goldberg, M.B., and Glogauer, M. (2016). Nuclear Factor Erythroid 2-Related Factor 2 Down-Regulation in Oral Neutrophils Is Associated with Periodontal Oxidative Damage and Severe Chronic Periodontitis. Am J Pathol *186*, 1417-1426.

Tavora, B., Mederer, T., Wessel, K.J., Ruffing, S., Sadjadi, M., Missmahl, M., Ostendorf, B.N., Liu, X., Kim, J.Y., Olsen, O.*, et al.* (2020). Tumoural activation of TLR3-SLIT2 axis in endothelium drives metastasis. Nature *586*, 299-304.

Thammavongsa, V., Missiakas, D.M., and Schneewind, O. (2013). Staphylococcus aureus degrades neutrophil extracellular traps to promote immune cell death. Science *342*, 863-866.

Thompson, A.A., Dickinson, R.S., Murphy, F., Thomson, J.P., Marriott, H.M., Tavares, A., Willson, J., Williams, L., Lewis, A., Mirchandani, A.*, et al.* (2017). Hypoxia determines survival outcomes of bacterial infection through HIF-1alpha dependent re-programming of leukocyte metabolism. Sci Immunol *2*.

Toepfner, N., Herold, C., Otto, O., Rosendahl, P., Jacobi, A., Krater, M., Stachele, J., Menschner, L., Herbig, M., Ciuffreda, L.*, et al.* (2018). Detection of human disease conditions by single-cell morpho-rheological phenotyping of blood. *ELife 7*.

Tole, S., Mukovozov, I.M., Huang, Y.W., Magalhaes, M.A., Yan, M., Crow, M.R., Liu, G.Y., Sun, C.X., Durocher, Y., Glogauer, M.*, et al.* (2009). The axonal repellent, Slit2, inhibits directional migration of circulating neutrophils. J Leukoc Biol *86*, 1403-1415.

Wagstaff, L., Goschorska, M., Kozyrska, K., Duclos, G., Kucinski, I., Chessel, A., Hampton-O'Neil, L., Bradshaw, C.R., Allen, G.E., Rawlins, E.L.*, et al.* (2016). Mechanical cell competition kills cells via induction of lethal p53 levels. Nat Commun *7*, 11373.

Wan, T., Zhao, Y., Fan, F., Hu, R., and Jin, X. (2017). Dexamethasone Inhibits S. aureus-Induced Neutrophil Extracellular Pathogen-Killing Mechanism, Possibly through Toll-Like Receptor Regulation. Front Immunol *8*, 60.

Werth, N., Beerlage, C., Rosenberger, C., Yazdi, A.S., Edelmann, M., Amr, A., Bernhardt, W., von Eiff, C., Becker, K., Schafer, A.*, et al.* (2010). Activation of hypoxia inducible factor 1 is a general phenomenon in infections with human pathogens. PLoS One *5*, e11576.

Ye, B.Q., Geng, Z.H., Ma, L., and Geng, J.G. (2010). Slit2 regulates attractive eosinophil and repulsive neutrophil chemotaxis through differential srGAP1 expression during lung inflammation. J Immunol *185*, 6294-6305.

Zhang, W., Lin, Y., Zong, Y., Ma, X., Jiang, C., Shan, H., Xia, W., Yin, L., Wang, N., Zhou, L.*, et al.* (2022). Staphylococcus aureus Infection Initiates Hypoxia-Mediated Transforming Growth Factor-beta1 Upregulation to Trigger Osteomyelitis. mSystems *7*, e0038022.

Zhou, S.L., Luo, C.B., Song, C.L., Zhou, Z.J., Xin, H.Y., Hu, Z.Q., Sun, R.Q., Fan, J., and Zhou, J. (2022). Genomic evolution and the impact of SLIT2 mutation in relapsed intrahepatic cholangiocarcinoma. Hepatology *75*, 831-846.